https://doi.org/10.5194/egusphere-2025-352



# Modeling the impact of drainage on peatland $CO_2$ and $CH_4$ fluxes and its underlying drivers

Thu Hang Nguyen[1], Philippe Ciais[1], Liyang Liu[1], Yi Xi[1], Chunjing Qiu[2], Elodie Salmon[1], Aram Kalhori[3,] Christophe Guimbaud[4], Matthias Peichl[5], Joshua L. Ratcliffe[5,6], Koffi Dodji Noumonvi[5], Xuefei Li[7]

[1] Laboratoire des Sciences du Climat et de l'Environnement, IPSL, CEA-CNRS-UVSQ, Université Paris-Saclay, 91191 Gif sur Yvette, France
[2] Research Center for Global Change and Ecological Forecasting, School of Ecological and Environmental Sciences, East China Normal University, Shanghai, China
[3] GFZ Helmholtz Centre for Geosciences, 14473, Potsdam, Germany
[4] Laboratoire de Physique et de Chimie de l'Environnement et de l'Espace, LPC2E, CNRS, OSUC, Univ Orleans, F-45071 Orléans, France.
[5] Department of Forest Ecology and Management, Swedish University of Agricultural Sciences, 90183 Umeå, Sweden
[6] Unit for Field-Based Forest Research, Swedish University of Agricultural Sciences, 92291 Vindeln, Sweden
[7] Institute for Atmospheric and Earth System Research/Physics, Faculty of Science, University of Helsinki, 00014 Helsinki, Finland

*Correspondence to*: Philippe Ciais (philippe.ciais@lsce.ipsl.fr)

**Abstract.** Peatland drying is an important process affecting greenhouse gas (GHG) emissions. Ditching for forest drainage has been standard forest management practice in the Nordic countries in the past centuries, and drying increasingly occurs also from climate change induced drought. Previously published meta-analyses from literature suggest that typically, drainage increases $CO_2$ emissions by enhancing oxic decomposition in aerated upper layers while suppressing $CH_4$ emissions. However, the data do not elucidate short term variations of GHG fluxes during drainage and usually only regress GHG emissions as a function of the annual mean water table. Here we developed a new parameterization of drainage in a land surface model that represents peat processes and fluxes of $CO_2$ and $CH_4$, by adding a machine-learning module to predict the daily water table depths from simulated soil moisture in the upper soil layers and a ditch which receives drainage water. Because peatland pre-drainage GHG emissions differ between sites and influence subsequent changes from drainage, the simulations are performed for virtual drainage applied to a collection of 10 pristine sites at which the model parameters are calibrated against observed GHG fluxes. Different drainage intensities are simulated by prescribing lower water table depths from the ditch depth, from 5 to 50 cm below the initial water surface. The resulting GHG flux changes across sites are compared with meta-analysis data from northern sites and show realistic results with a reduced $CO_2$ sink and reduced $CH_4$ emissions. Additional comparison with continuous flux data collected in the UK for different sites associated with increasing drainage levels also shows good model performances. Overall, using GWP100 to compare the effect of $CH_4$ vs. $CO_2$ flux changes, our simulation results suggest only very small net GHG emission changes when $CH_4$ is expressed in $CO_2$-equivalents units using GWP100, when peatland is drained for 50 years, yet with differences between sites. Over time during 50 years of drainage, the emission factors of $CO_2$ flux decrease because of exhaustion of labile soil organic substrate for





decomposition and the reduction of $CH_4$ emissions is amplified, also because of less material for anoxic decomposition. The sensitivities of $CO_2$ flux changes to increased water table depth changes are primarily controlled by initial $CO_2$ and $CH_4$ fluxes, initial soil carbon content, peat vegetation community, air temperature and initial water table depth. The influence of peat vegetation on the GHG flux sensitivities in the model occurs via differing lability of soil organic carbon pools, with

moss-dominated sites having a lower sensitivity due to their longer peat turnover time. Our calibrated process-oriented model simulations of the sensitivities of GHG flux changes to water table depth can be emulated by linear regression models, which are simple and could be used in decision support tools and GHG regional budgets accounting.

## 1 Introduction

In peatlands, organic mater does not fully decompose due primarilly to the lack of oxygen in the soil caused by waterlogged

conditions. Peatlands have therefore accumulated a large amount of carbon over time, ~450Gt (Joosten, 2009; Page et al., 2011), though they also emit a significant amount of methane ($CH_4$) during anaerobic decomposition (commonly 5–80 mg $m^{-2}$ $day^{-1}$ in northern peatlands according to Blodau (2002). However, since the last few centuries, drainage of peatlands has become widespread for various purposes, such as commercial forestry (Arnold et al., 2005), farming (Kiew et al., 2020; Qiu et al., 2021), livestock grazing (Conchedda and Tubiello, 2020; Nieveen et al., 2005), and peat extraction for fuel (Schilstra

and Gerding, 2004; Sirin et al., 2010). When peatlands are drained, increased oxygen levels in the soil promote aerobic decomposition of organic matter, leading to higher $CO_2$ emissions into the atmosphere and reducing $CH_4$ emissions (Holden, 2005; Rydin and Jeglum, 2013; Evans et al., 2021).

Field studies have been conducted to assess the impact of recent drainage on peatlands' contemporary greenhouse gas

(GHG) emissions. Monitoring of recently drained sites using chambers (e.g., Furukawa et al., 2005; Laine et al., 2009; Martikainen et al., 1995; Munir et al., 2015; Strack and Waddington, 2007; Swails et al., 2022) and eddy covariance flux towers (Tikkasalo et al., 2024; Tong et al., 2024) have been performed to measure GHG fluxes, but the amount of data are still scarce, limited to few sites and short term duration (mostly < 10 years). When a site is being monitored which has been drained for a century, contemporary monitoring data observe the effect of drainage in the present, but as a consequence of

biophysical and biochemical changes in the past. Apart from such contemporary diachronic observations of GHG fluxes at sites that have been drained in the past, only few studies based on peat coring have actually been able to assess the impact of historic drainage.

In addition to the primary literature, meta-analyses gathering data from various sites and methods (Couwenberg et al., 2010;

Maljanen et al., 2010; Prananto et al., 2020) have been combined with empirical upscaling models of drainage-induced GHG fluxes (Evans et al., 2021; Huang et al., 2021b) across diverse geographic regions, peatland types and conditions. Some meta-analyses suggest that post-drainage peat decomposition causes long-term legacy CO2 emissions, decades after drainage



(Couwenberg et al., 2010; Huang et al., 2021b; Zou et al., 2022). Concurrently, meta-analyses show that $CH_4$ emissions are reduced and $N_2O$ emissions can be increased during drainage (Zou et al., 2022; Tikkasalo et al., 2024), the later of which

applies to N-rich sites.

Previous studies also showed that water table level (WTL) under drained conditions controls the response of emissions, but there may be distinct WTL thresholds for each gas and for each peatland. For instance, Zou et al. (2022) found that $CO_2$ and $N_2O$ emissions only increase significantly when WTL is deepened by more than 30 cm below the soil surface. On the other

hand $CH_4$ emissions are suppressed when WTL is deepened to 5 cm below the soil surface, and then remain low for deeper WTL. Overall, using 100-year Global Warming Potential (GWP100) across multiple sites to compare $CH_4$ and $N_2O$ with $CO_2$, Zou et al. (2022) showed that net $CO_2$-equivalent emissions are reduced by drainage when the WTL depth drops from being above the surface to 5 cm below the soil surface because of the predominant role of $CH_4$ reductions. When the WTL depth is below this typical threshold, net $CO_2$-equivalent emissions can increase with drainage from the increase of $CO_2$

emissions offsetting the reduction of $CH_4$ emissions. However, those meta-analysis studies do not separate well site-specific responses, do not describe systematic effects of different initial conditions and initial water table depth during drainage. Due to the dominance of short-term drained experiments, data from meta-analysis do not offer a complete assessment of net climate effects caused by distinct decadal changes of $CO_2$, $CH_4$ and $N_2O$ fluxes.

Another approach involves the use of process-based models, which can simulate the processes co-controlling $CH_4$ and $CO_2$ emissions by coupling water, thermal, and greenhouse gas biochemistry processes and drainage (Huang et al., 2021a; Kwon et al., 2022). Models are versatile, capable of simulating at various time scales and can be tailored to specific site conditions, though their reliance on generic parameterizations means they cannot always be precisely calibrated for individual sites (Liu et al., submitted). In addition, process-based models tend to be complex, and interpreting their results can therefore be

challenging. Although process-based models have been increasingly applied to studying peatland dynamics over the past decade (Mozafari et al., 2023), the majority of existing models were not specifically designed for peatlands (Mozafari et al., 2023), and their use in simulating peatland drainage remains relatively limited. Although few peat-enabled land surface models have been applied at sites and over large regions using gridded simulations (Qiu et al., 2021) to study GHG fluxes of northern peatlands in response to climate changes and rising atmospheric $CO_2$, the impact of drainage on GHG emissions has

not been fully explored with these models. For instance, Qiu et al. (2021) used the ORCHIDEE-PEAT model to simulate peatland $CO_2$ flux changes for the historical conversion of northern peatlands to croplands and found a large cumulative loss of 70 PgC, but they prescribed to their model direct changes from peatland to cropland instead of using a transient drainage period when the WTL is lowered and the peatland becomes altered for instance with soil compaction, followed by crop or pasture cultivation. Further, they did not study the effect of $CH_4$ emissions reduction after conversion to agriculture. Kwon et

al., (2022) used a version of the same model that include the $CH_4$ cycle (Salmon et al., 2022) for simulating the effect of drier conditions at six Arctic peatland sites and found that lowering the water level by 10 cm reduced the $CO_2$ sink by $13 \pm 15$ g C





m$^{-2}$ year$^{-1}$ and decreased $CH_4$ emissions by $4 \pm 4$ g $CH_4$ m$^{-2}$ year$^{-1}$ leading to reduced accumulation of carbon over the next hundred years. For temperate peat sites, Evans et al. (2021) further showed that long term water table drawdowns at most sites led to net carbon loss.


To our knowledge, there has not been any systematic modeling of the effect on GHG emissions of anthropogenic drainage of peatlands using land surface models that can simulate both $CO_2$ and $CH_4$ fluxes. One difficulty is that climate and pre-drainage fluxes and water levels are generally not measured continuously at sites subject to drainage, which prevents the calibration of models before the perturbation. In this study, we use the ORCHIDEE-PEAT, a land surface model developed

to include peat processes for $CO_2$ and $CH_4$ fluxes to address this research gap. Our strategy is to model virtual drainage conditions at 10 real pristine peatland sites in temperate and boreal regions (one in the US, one in Canada, one in Japan, others in Europe) which are not currently subjected to intentional drainage using artificial ditches. All the sites have continuous $CO_2$ observations from eddy covariance and 8 sites also have $CH_4$ flux observations from flux chambers. All the sites have continuous local hourly climate data used as model input and local water table depth measurements are also

available at most sites. This set of well-observed sites allows us to calibrate the model before drainage. The changes of $CO_2$ and $CH_4$ emissions for different drainage intensities will be simulated by prescribing increasingly deeper water table depths. These drainage simulations will then be evaluated against meta analysis results from northern temperate and boreal peatland sites (Huang et al., 2021b; Zou et al., 2022) and against a detailed set of flux measurements conducted across the UK by Evans et al. (2021) for different water table depths.


We performed site simulations with virtual drainage under current climate conditions, with different prescribed water table depths, to address the following questions: (1) what are the changes of $CO_2$ and $CH_4$ fluxes in response to drainage if we would drain even more peatlands and how do they compare with observations, (2) how do fluxes change as a function of drainage duration, (3) what is the modeled sensitivity of flux changes to water table depth, (4) what factors affect the

sensitivity at each site in the model, (5) what is the net climate effect of $CO_2$ and $CH_4$ flux changes induced by drainage using the GWP100 metrics to compare $CH_4$ and with $CO_2$. There are other metrics such as GWP* (Lynch et al., 2020) and SGWP (Neubauer and Megonigal, 2015), but we use here the GWP100 as it is used by the UNFCCC and national inventories for comparing the two gases. In the following, we present the model and its modifications for simulating drainage (Section 2), the model performances for simulating $CO_2$ and $CH_4$ fluxes and water table depth before drainage, and

the results of changes in fluxes during the drainage phase (Section 3). First, we compare the modeled flux changes to meta-analysis results and field measurements to evaluate the model results, to ensure that they fall within plausible ranges. Then, we simulate emission factors defined by the increase of $CO_2$ emissions and the decrease of $CH_4$ emissions over time for various durations of drainage. Finally, we analyze the sensitivities of $CO_2$ and $CH_4$ flux changes per unit of water table deepening in comparison with independent observations and analyze the factors that explain why the model has different



sensitivities between sites. A discussion of our results and perspectives for future large scale simulations is given in Section 4.

## 2. Methods

### 2.1 Peatland enabled land surface model

ORCHIDEE-PEAT is a land surface model (https://orchidee.ipsl.fr/) with a module specifically developed for peatland. In
general, the model comprises two main components: (1) Energy and water balance, and (2) Vegetation and soil carbon cycle. In each model grid cell defining the spatial unit of simulation, the water balance is simulated individually for each soil tile containing a different type of vegetation, while the energy balance is simulated for the whole grid cell (Xi et al., 2024). The soil hydrology is simulated using 11 vertical layers, considering incoming rainfall or snowfall, soil water evaporation, water infiltration in the soil profile every half-hour, surface runoff and drainage (also called subsurface runoff in similar models).
One soil tile is specifically designated for peat with distinct soil hydrological parameters. To ensure that this peat soil tile keeps a high water content, bottom drainage is excluded and surface runoff from non-peat soil tiles is given to peat soil tile at each time step, with a slab water layer that can be created above the peat soil surface of maximum thickness 10 cm. The vegetation and soil carbon cycle component calculates biogeochemical and biophysic variables for each plant function type (called PFT). A peatland tile can contain one or more specific PFTs that grow on it, with the possibility to prescribe a
fraction of moss, graminoids (= grasses + sedges) and shrubs.

The peatland percentage at a site is defined by the total fraction of peatland PFTs in the grid cell where the site is located. A more detailed description about the model can be found in Qiu et al. (2018) and Qiu et al. (2019). In addition, an improved routine for methane simulation for peatland was integrated into the model by Salmon et al. (2022). This enables this study to
be concerned with $CO_2$ fluxes and $CH_4$ emission. The $CO_2$ flux is represented by Net Ecosystem Exchange (NEE) which is calculated as the algebraic sum of gross primary productivity (GPP, negative sign), autotrophic respiration (AR, positive sign), and heterotrophic respiration (HR, positive sign). Methane produced in the soil layers is transported to the soil surface via plant-mediated transport, ebullition, and diffusion (positive sign).

Meteorological forcings used as input for the model (including precipitation, air temperature, air humidity, pressure, solar radiation, and winspeed) are extracted from the collection of CRU JRA forcing datasets (Friedlingstein et al., 2022). The 6-hour time resolution of CRU JRA data is automatically interpolated to 30 minutes (default time step) by the ORCHIDEE model.





## 2.2 Site description and simulation protocol

The drainage simulations will be implemented for 10 peatland sites in temperate and boreal regions briefly described in Table A1 (Appendix A). The vegetation composition (graminoids, shrubs, and mosses) is taken from the literature when available; otherwise, it is estimated visually from satellite images.

The ORCHIDEE-PEAT model was first run for 100 years to reach the equilibrium state of soil thermal and hydrological
conditions, followed by 10,000 years of spin-up to accumulate soil carbon with a fast module that computes only soil carbon changes from daily litter input and soil climate archived from the first simulation (Qiu et al., 2018), using repeated forcing data and a pre-industrial atmospheric $CO_2$ concentration of 285 ppm. An additional 100-year simulation was then conducted before flux observations (Table A1) to simulate biogeochemical processes under observed atmospheric $CO_2$ concentrations. The model was then calibrated for each site under present-day conditions following (Liu et al., submitted), here calibrating
the parameters controlling $CH_4$ fluxes in addition to $CO_2$ fluxes for sites that have $CH_4$ flux observation. For $CO_2$ fluxes, parameters governing photosynthesis rates, stomatal conductance, autotrophic respiration, soil organic carbon oxic decomposition rate and its sensitivity to temperature (Table B1 in Appendix B) were calibrated. For $CH_4$, we calibrated parameters driving the methane production, oxidation, and transport (Table B1). The calibration was performed using the ORCHIDEE data assimilation system (ORCHIDAS, Peylin et al., 2016), a Bayesian statistical framework, employing a
genetic algorithm to minimize a cost function between observation and simulated outcomes (Tarantola, 2005), i.e. here between observed and simulated daily NEE, and daily $CH_4$ emission simultaneously. For sites without $CH_4$ flux observations, only $CO_2$ parameters can be calibrated, while parameters related to $CH_4$ processes were assigned using the average calibrated values from all sites with available $CH_4$ flux observations.

Pre-drainage simulations were run for each site over a period of 100 years. We then simulate five virtual drainage scenarios for each site over a subsequent 50-year period in which the water table remains at its initial/baseline level and varies each day (no drainage), or is prescribed at 5, 10, 20, and 50 cm below the initial/baseline water level. For the 150 years of simulations, 6-hourly climate forcings of the flux-observed years are used by randomly selecting years. The same years are used for the undisturbed and drained simulations to ensure that the resulting differences in fluxes are not caused by
differences in climate.

## 2.2 Reconstruction of baseline water table

Performing drainage simulations requires a good representation of the baseline water table (WTD) before drainage. This is not straightforward in a model like ORCHIDEE where the numerical discretization of the soil into layers is coarser with increasing depth, e.g. a layer has thickness of 25 cm at 50 cm below the surface, which does not allow to position the water
table in this layer accurately. Therefore, we developed a machine learning module, separate from the ORCHIDEE model, to



simulate the accurate position of the water table as a function of simulated soil moisture in the soil layers. The module is trained to gap-fill the daily baseline water table observed at 8 sites and it is used to reproduce the missing water table measurements at 2 sites (CZ-Wet, DE-Akm; Table A1). Because the water table is closely related to soil moisture, our strategy is to extract the simulated soil moisture of all the layers to train a model defined by $WTD = f(SM)$. Simulated soil moisture and observed water table time series are sampled by 7-day blocks to make up the training dataset for the model, with soil moistures as inputs and water table as output. Firstly, for a site where observed water table data is available but with missing values, its own training dataset is divided into an 80/20 ratio for training and testing, then the missing data are filled. This process is referred to as self-reconstruction. Secondly, a multi-site training dataset is created using all the sites that have water table observation with a training/testing ratio of 50/50 for four sites in Sweden and 80/20 for the remaining. The contribution of each Swedish site to the total training subset is reduced due to their proximity to each other, which potentially results in the reproduction of their relationships between water table and soil moisture. Once this multi-site model has been trained, it will take the simulated soil moisture (from ORCHIDEE-PEAT) of water-table-unknown sites as input to derive the baseline water table for these sites as output. In this study, we used a Support Vector Regression (SVR) model (Smola and Schölkopf, 2004) to simulate the relationship between soil moisture and water table, because of its advantages such as robustness to outliers and effectiveness in high-dimensional space (Mohammed Rashid et al., 2022).

**2.3 Peatland drainage by ditching**

In order to simulate the effect of drainage on $CO_2$ and $CH_4$ fluxes, we assume that the water table is lowered by draining into ditches. We introduce into the model a new module with a ditch of which depth is the same as the desired lowered maximum water table depth. In addition, runoff from other soil tiles to peat is assumed to be excluded during drainage implementation, reducing water supply to peat. Note that the simulated water table fluctuates continuously throughout the year from variable incoming rainfall, and so does the ditch water level. The difference in the hydrology simulations with and without drainage is shown in Figure 1 and described below.

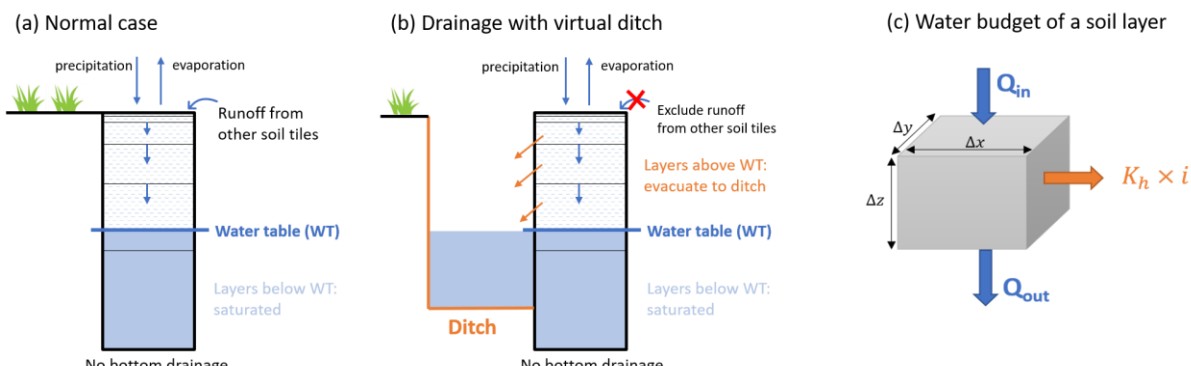



**Figure 1. Illustration of drainage by ditch simulation in the ORCHIDEE-PEAT model: (a) Soil column without a ditch, showing vertical water fluxes and horizontal delivery from the runoff of adjacent soil tiles without peat. (b) Soil column with a ditch, incorporating lateral water loss to the ditch (orange) and suppression of runoff from other tiles. (c) Water budget of a soil layer, with lateral output fluxes due to drainage into the ditch highlighted in orange. The dimension of the volume of soil is defined by $\Delta x, \Delta y$ and $\Delta z$. $Q_{in}$ and $Q_{out}$ are inward and outward water fluxes and $K_h$ is the horizontal hydraulic conductivity and $i$ is the hydraulic gradient.**

In the ORCHIDEE model, each soil column is separated into layers. We set soil layers below the water table to be saturated ($\theta=\theta_s$). Above the water table, in a volume of soil with horizontal dimension $\Delta x, \Delta y$ and thickness $\Delta z$, without drainage, water flows through the soil in the vertical dimension from the infiltration of rainfall events and runoff given by other soil tiles in the same grid (Fig. 1(a), blue arrows), a flux $Q_{in}$ entering and a flux $Q_{out}$ leaving the layer across the cross-section area $\Delta x \Delta y$ (Fig. 1(c)). These vertical fluxes are functions of hydraulic conductivity and diffusivity ($K_v$ [m s$^{-1}$] and $D_v$ [m$^2$ s$^{-1}$], respectively, with v for vertical) which are functions of soil moisture ($\theta^{t+\Delta t}$). With drainage, a new horizontal flux across the cross-section area $\Delta y \Delta z$ is added for representing water running out of the soil to the ditch (Fig. 1(c), orange arrow). Assuming that ditches are placed perpendicularly to the flow of groundwater, this additional drainage flux is a function of horizontal hydraulic conductivity ($K_h$) and hydraulic gradient ($i$). The change of moisture in a time step $\Delta t$ is given by:

$$\frac{\Delta \theta}{\Delta t} \Delta x \Delta y \Delta z = (Q_{in} - Q_{out}) \Delta x \, \Delta y - (K_h \, i) \, \Delta y \, \Delta z \qquad (1)$$

Removing $\Delta x \Delta y$ in both sides, Eq. (1) becomes:

$$\frac{\Delta \theta}{\Delta t} \Delta z = (Q_{in} - Q_{out}) - (K_h \, i) \frac{\Delta z}{\Delta x} \qquad (2)$$

with $\Delta x$ now playing a role as ditch spacing. Using a numerical method with boundary conditions at top layer where ($Q_{in}$ = (precipitation - soil evaporation) and at bottom layer where $Q_{out}$ = free drainage = zero, a tridiagonal matrix system is constructed to solve $\theta^{t+\Delta t}$ (Ducharne et al., 2018).

## 3 Results

### 3.1 Simulation before drainage

The results of the model calibration under present condition are shown in Fig. 2, with $CO_2$ fluxes (Net Ecosystem Exchange, NEE) and $CH_4$ emissions calibration for sites that have observations of both fluxes (a-h), and $CO_2$-only calibration for other sites (i-j). The $CH_4$ fluxes of all eight sites were well simulated, with RMSE ranging from 0.016 to 0.024 g $CH_4$ m$^{-2}$ day$^{-1}$. The calibrations of NEE also showed good performance, with RMSE smallest at Sweden sites (from 0.574 to 0.8 g $CO_2$ m$^{-2}$ day$^{-1}$).





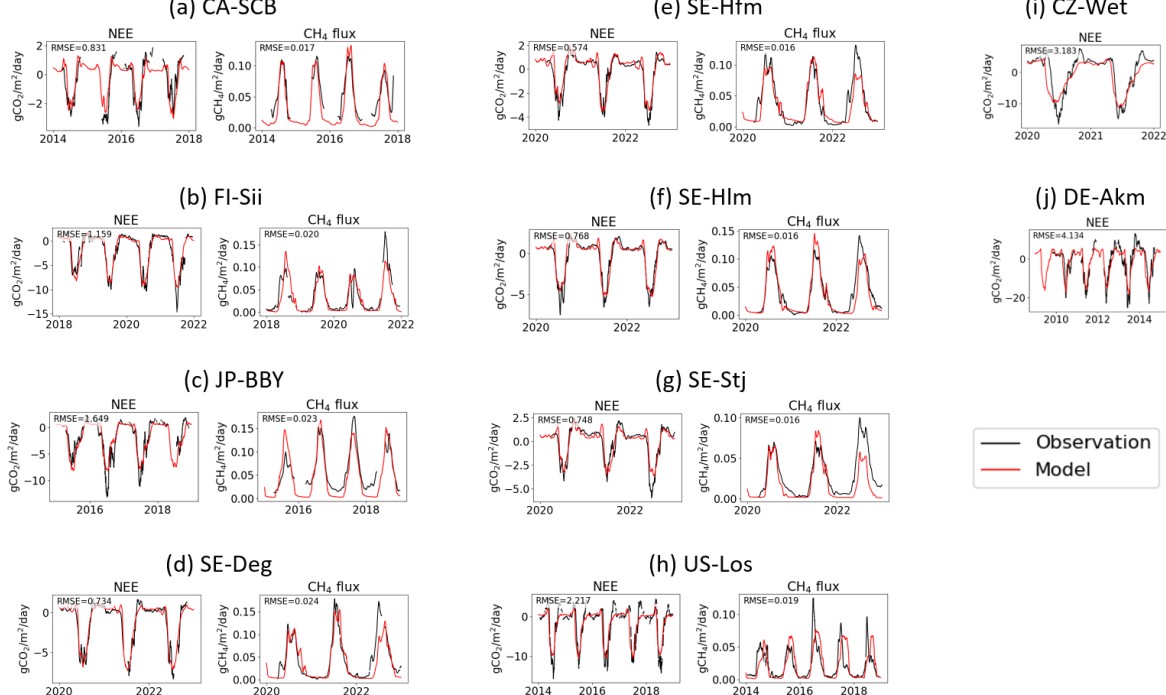

**Figure 2: Comparison of model simulations after calibration with observed net ecosystem exchange CO$_2$ fluxes (NEE) and CH$_4$ emissions. Sites in panels a-h measured both fluxes and in i-j only NEE.**

Figure 3 presents the reconstruction of water table depth from soil moisture with the SVR model trained for individual sites (Fig. 3 a-h, left column). The RMSE of modelled and observed water table depth ranged from 2.76 cm (FI-Sii) to 6.11 cm (US-Los) and the model captured in general the seasonal variations very well, but less so the short-term variations. At the Swedish sites, the peat is frozen between mid-October to early May, during which time the WTD is not fluctuating in reality. However, due to the lack of WTD measurements during these frozen periods, the self-reconstructed WTD from the SVR model for these times were accepted as the best available approximation.

With the SVR model trained using all the sites together, SVR was less effective because it must compromise the patterns between all the sites: RMSE increased up to 6.76cm but still captures the main seasonal variations. The water table depth reconstructed using SVR trained from all other sites and applied at the two sites where direct WTD observations were missing including CZ-Wet, DE-Akm (Fig. 3 i-j, right column) was used in this study as the baseline water table depth.







**Figure 3. Water table depth reconstruction with a machine learning model trained on observations and using modelled soil moisture in all soil layers within the top 2 meters below surface (red) compared with observed values (black).**

## 3.2 Drainage simulation: changes of $CO_2$, $CH_4$ fluxes, soil water and oxygen after few years

Drainage reduced the water content in the soil and increased oxygen concentration (Fig. 4). With more oxygen in the soil and less moisture, heterotrophic respiration was increased. On the contrary, methane production was suppressed in aerated soil,





leading to less methane emission when water table gets lower. Figure 4 presents the monthly average of soil water content, oxygen, NEE and CH$_4$ emissions during the first three years after drainage, for different WTD levels. It shows that the

inclusion of a ditch in the model could lower soil moisture effectively and that flux simulation responded as expected with decreased NEE uptake or NEE switching to a net CO$_2$ source, and decreased CH$_4$ emissions. Note that there could be a slight inconsistency between the pre- and during-drainage simulations. In the pre-drained simulation, soil moisture was calculated by the model, whereas in the drainage simulation, all the soil column below the ditch depth was forced to be saturated, which introduces a different water content of the deep layers. For instance, a deep layer that was near-saturated in the pre-drainage

simulation was set to become saturated in the drainage simulation. The impact of this inconsistency on gas fluxes, however, was small because deep layers did not contribute much changes in CH$_4$ and CO$_2$ fluxes (they contain less labile organic carbon) and they were already near-saturation or at full saturation before drainage. The 50-year time series of NEE and CH4 emission are shown in Figure C0 (Appendix C).

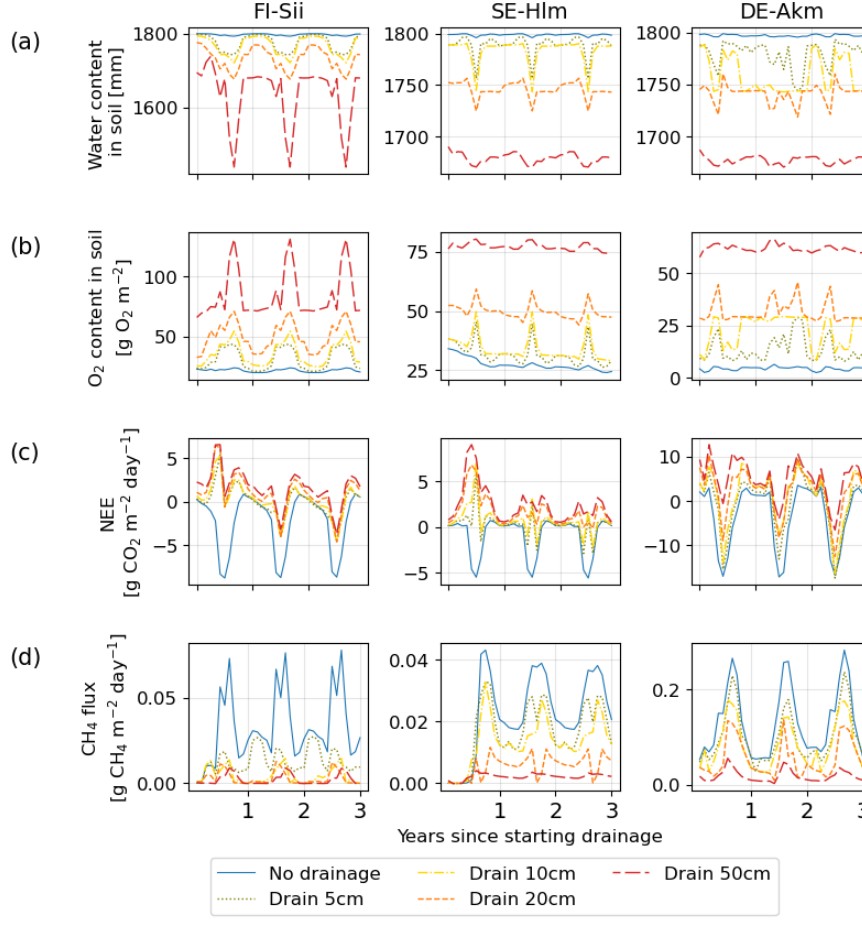






**Figure 4. Simulated changes at three sites of (a) soil water content in the whole profile (2m), (b) soil oxygen content, (c) Net Ecosystem Exchange of $CO_2$ and (d) methane emissions. Different colors indicate different water table depths below the original water level of each site.**

### 3.3 Long-term changes of $CO_2$ and $CH_4$ fluxes, diagnosed by emission factors after up to 50 years

Figure 5 shows the average emission factor (EF) across all peatland sites for 50 years since starting drainage. The EF is defined by the average GHG flux per unit of area over a given period of time under drainage minus the flux at the same site under undisturbed conditions. Qiu et al. (2021) implemented similar calculations using the ORCHIDEE-PEAT model for estimating carbon emission factors after peatland cultivation. Their emission factors calculated over two decades following conversion of peatland to cropland was 16 g $CO_2$ $m^{-2}$ $day^{-1}$ as a median, ranging from 5 to more than 50 g $CO_2$ $m^{-2}$ $day^{-1}$. Our

estimates are lower than their values, with a median of 1.88 g $CO_2$ $m^{-2}$ $day^{-1}$ (95%CI: 0.21 to 8.65), and a few drained sites show a negative EF (0.08% of all sites-months in the first two decades when drained 50cm), i.e., a slightly stronger $CO_2$ sink when drained because deep soil layers remain saturated and continue to accumulate carbon (Fig. 5). Our lower median EF values than Qiu et al. (2021) are because deep soil layers remain saturated in our simulations, preserving their carbon from decomposition, while Qiu et al. (2021) assumed that the entire soil profile became drained and under-saturated when

converting to cropland.

Regarding the temporal variation of emission factors, we found a decrease in $CO_2$ EF between the first and the fifth decade (Fig. 5) from 3.14 g $CO_2$ $m^{-2}$ $day^{-1}$ (0.23 to 10.97) down to 1.18 g $CO_2$ $m^{-2}$ $day^{-1}$ (0.15 to 3.19) for 50 cm drainage, a similar trend than the findings of Qiu et al. (2021). As GPP and autotrophic respiration (AR) was found to be little affected during

the drainage period (Fig. C1-f,g), the change in emission factor over time was primarily driven by the increase of heterotrophic respiration (HR) under drainage compared to undrained conditions (Fig. C1-e). HR includes one part from litter and one from soil. In the early decade, the decomposition of litter accelerated in case of drainage, leading to the quick decrease in litter amount (Fig. C1-a). Later, the reduced availability of litter for decomposition caused the heterotrophic respiration from litter in drained peatlands to approach the same levels than in undrained peatlands (Fig. C1-b) where litter

accumulated gradually, or decomposed at a much slower rate, and maintained a stable and low heterotrophic respiration rate. On the other hand, while SOC in undrained peatlands accumulated gradually, SOC in drained peatlands increased rapidly in the early decades due to a large input of carbon from litter but then slowed down as litter carbon input declined (Fig. C1-c). Consequently, the difference in soil heterotrophic respiration (HR) between drained and undrained peatlands narrowed over time (Fig. C1-d). These declining trends of both soil and litter HR in drained peatlands explains why the simulated EF

decreased by time. A small decreasing trend in EF over time following drainage was determined experimentally by Truskavetskii (2014) based on soil measurements from chronosequence data. Rojstaczer and Deverel (1993) reported a trend in the subsidence of organic soil over time using periodic leveling surveys, suggesting decreased EF as well, although our model did not explicitly include subsidence and compaction effects. For $CH_4$ emissions, on the contrary, the decrease of EF amplifies with time, going from -0.06 g $CH_4$ $m^{-2}$ $day^{-1}$ in the first decade down to -0.09 g $CH_4$ $m^{-2}$ $day^{-1}$ in the last decade in





case of 50 cm drainage as shown in Fig. 5 (negative EF in this Figure indicating less $CH_4$ emissions in the drained than in the undrained).

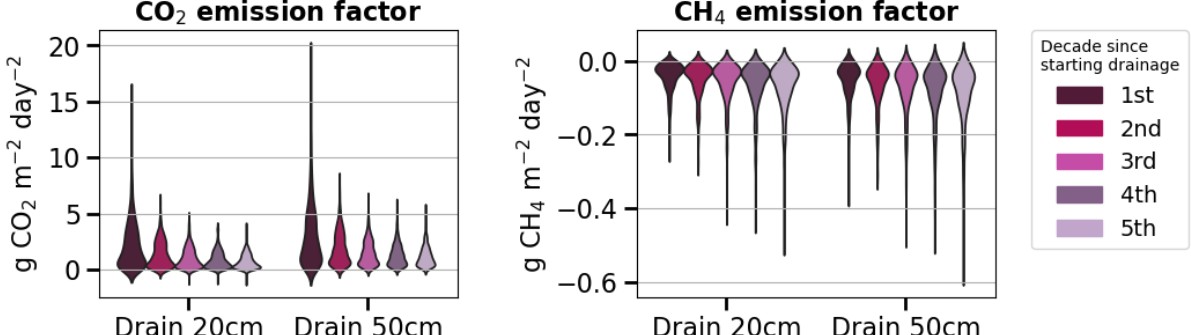

**Figure 5: Evolution of modelled emission factors (a) for $CO_2$ fluxes and (b) for CH4 emissions for five decades after drainage, displayed for two drainage depths**

### 3.4 Comparison of simulated greenhouse gas flux changes with observations

To evaluate our drainage simulation results for GHG fluxes, we used observed changes of $CO_2$ and $CH_4$ fluxes collected from multiple sources through two meta-analyses of drained wetlands experiments (Huang et al., 2021b; Zou et al., 2022), the network of field flux measurements collected in Europe by Evans et al. (2021), and measurement collected in the dataset

of          Ecosystem          Services          publicly          provided          by          the          WET          HORIZONS          project (https://www.wethorizons.eu/resources/#database). Water table depths were provided along with the fluxes in both meta-analysis studies, while in Evans et al. (2021), an 'effective' water table depth was used, defined as the smaller value between the measured water table depth and the measured peat depth. The data collected by Huang et al., (2021b) provided measurements of undisturbed fluxes at control sites, as well as 77 measurements of fluxes after < 1 year, 42 after 1-10 years,

and 78 after > 10 years of drainage. Since the impact of drainage on NEE was shown to be larger in the first decade after the start of drainage (Fig. 5), we compared 10-years average values of our simulation with these data. Figure 6a,b shows the distribution of simulated and observed NEE and $CH_4$ fluxes for different water table levels, regardless of the site and drainage level. Overall, our simulation results lie within the large range of the observed data, although our modeled $CH_4$ emissions are in the upper range of the observed distribution. In Evans et al. (2021), a linear relationship was obtained

between effective water table depth and net ecosystem production (NEP, the sum of NEE and biomass removal). Retaining only sites within boreal and temperate zones from their study (the yellow points in Fig 6a,b), we confirm such a strong linear correlation between effective WTD and NEE ($R^2=0.66$, $p<0.001$). Our results (the red points in Fig 6a,b) were in good agreement with Evans et al. (2021), yet a weaker correlation between WTD and NEE was obtained in the model ($R^2=0.59$, $p<0.001$). Meta-analysis data mixing different sites, experiment duration and conditions showed a weak correlation ($R^2 =$



0.18 in Huang et al. (2021b), and $R^2$ = 0.04 in Zou et al. (2022), suggesting that the Evans et al. (2021) field measurements with continuous flux and WTD records at the same site may be a better benchmark of the model results.

The distributions of the changes of fluxes compared to no drainage are presented in Figure 6c,d for different drainage levels (ΔWTD). For this comparison, we kept only samples for 5, 10, 20, 50 cm (± 10%) drainage levels in the reference data. The
numbers of measurements for each drainage level differs and different drainage levels can include different peatland sites and types. The results indicate that the modeled distribution of ΔNEE falls within the observed distributions from Huang et al. (2021b) and Zou et al. (2022), although all of our simulations show positive ΔNEE values ranging from 0 to 5.63 g $CO_2$ $m^{-2}$ $day^{-1}$, i.e., more $CO_2$ is emitted or less $CO_2$ absorbed after peatlands are drained, whereas the meta-analysis data exhibit either increases or decreases of NEE (-7.33 to 17.80 g $CO_2$ $m^{-2}$ $day^{-1}$ in Huang et al. (2021b), and -11.75 to 10.39 g $CO_2$ $m^{-2}$
$day^{-1}$ in Zou et al. (2022). For methane, all ΔCH4 values are negative in our model, ranging from -0.1 to 0 g $CH_4$ $m^{-2}$ $day^{-1}$, i.e. less $CH_4$ is emitted when peat is drained, while the meta-analysis data shows a negative median value of the distribution, yet with negative or positive changes of ΔCH4 (-0.16 to 0.22 g $CH_4$ $m^{-2}$ $day^{-1}$ in Huang et al. (2021b); -1.03 to 0.40 g $CH_4$ $m^{-2}$ $day^{-1}$ in Zou et al. (2022). These discrepancies mainly come from the fact that the meta-analysis studies took into account emission variations for different WTDs which can come from natural climate, microtopographic conditions and laboratory
experiments, while our study focuses on ditch drainage conditions. For example, Huang et al. (2021b) included a comparison between a lawn with near-surface WTD and an adjacent hollow filled with water, which gave a negative ΔNEE because the hollows leached dissolved or particulate organic carbon, facilitating oxidation, thus releasing more $CO_2$ than the lawn (Villa et al., 2019). A decrease in NEE may also result from anomalously cold weather, which can reduce respiration more significantly at the drained site than at the undrained site (Renou-Wilson et al., 2016), given that dry soil is more susceptible
to air temperature fluctuations, and low temperatures inhibit soil decomposition. Additionally, inconsistencies in the data may contribute to their large range of ΔNEE, as they include different drainage levels and varying durations. Potential errors in the meta-analysis studies, such as inconsistencies in data collection, may also play a role in estimating negative ΔNEE or positive ΔCH4; for instance, our inspection of meta-analysis data found cases where emission data reported for different depths were from different locations.








**Figure 6.** Simulated fluxes of (a) Net ecosystem exchange of $CO_2$ and (b) $CH_4$ emissions for different water table levels below the original water surface (red points) compared with meta-analysis observations from Huang et al. (2021b) (cyan) and Zou et al. (2022) (brown). The darkest color shading is one standard deviation of the observations, and the lightest shading is the min-max range. (c) Modeled (red dots) and observed distributions of NEE changes (ΔNEE) for different water table depths of drainage, compared with meta-analysis results, and (d) same for $CH_4$ emissions changes (ΔCH$_4$).

### 3.5 Sensitivity of GHG emissions changes to lowered water table depth

In this section, we analyzed the *sensitivity* of GHG fluxes defined by the change of NEE or $CH_4$ emission per cm of lowered WTD computed from our simulations, for different starting WTD levels at the same site, after 10 years of drainage. For the observations, we calculated the sensitivities from changes in GHG emissions reported by Huang et al. (2021b) and Zou et al. (2022) and changes of WTD between 5 and 50 cm. The distributions of the resulting sensitivities are shown in Fig. C2. Averaged across all sites, the modeled mean NEE sensitivity (0.09 g $CO_2$ m$^{-2}$ day$^{-1}$ cm$^{-1}$; 95%CI: 0.02 to 0.30) is close to the observation-based values derived from the data of Huang et al. (2021b); 0.12 g $CO_2$ m$^{-2}$ day$^{-1}$ cm$^{-1}$; -0.23 to 0.67), but higher than the data of Zou et al. (2022), although within their 95% confidence interval (0.01 g $CO_2$ m$^{-2}$ day$^{-1}$ cm$^{-1}$; -0.37 to 0.48). The simulated methane emission sensitivity (-1.58 mg $CH_4$ m$^{-2}$ day$^{-1}$ cm$^{-1}$; -4.5 to 0.03) is also comparable to the empirical



values (-1.93 mg $CH_4$ m$^{-2}$ day$^{-1}$ cm$^{-1}$; -15.96 to 0.21 of Huang et al. (2021b) and -1.70 mg $CH_4$ m$^{-2}$ day$^{-1}$ cm$^{-1}$; -19.82 to 5.10 of Zou et al. (2022). Kwon et al. (2022) used ORCHIDEE for 100-years drainage simulations at six arctic peatland sites with WTD lowered by 5 to 50 cm and reported average sensitivities of 0.01 +- 0.02 g $CO_2$ m$^{-2}$ day$^{-1}$ cm$^{-1}$ for NEE, and of -1.10 +-
1.10 mg $CH_4$ m$^{-2}$ day$^{-1}$ cm$^{-1}$ for $CH_4$ emissions. Our modeled NEE sensitivity is more positive and our modeled $CH_4$ emission sensitivity is slightly more negative than Kwon et al. (2022), despite using almost the same model. This may be attributed to the fact that we did simulation across a larger range of climate while Kwon et al. (2022) only simulated drainage of arctic wet peatlands where winter respiration is absent and high rainfall and snowmelt inputs maintain relatively more stable hydrological conditions, thus their NEE and $CH_4$ emissions being less sensitive to lowered WTD.


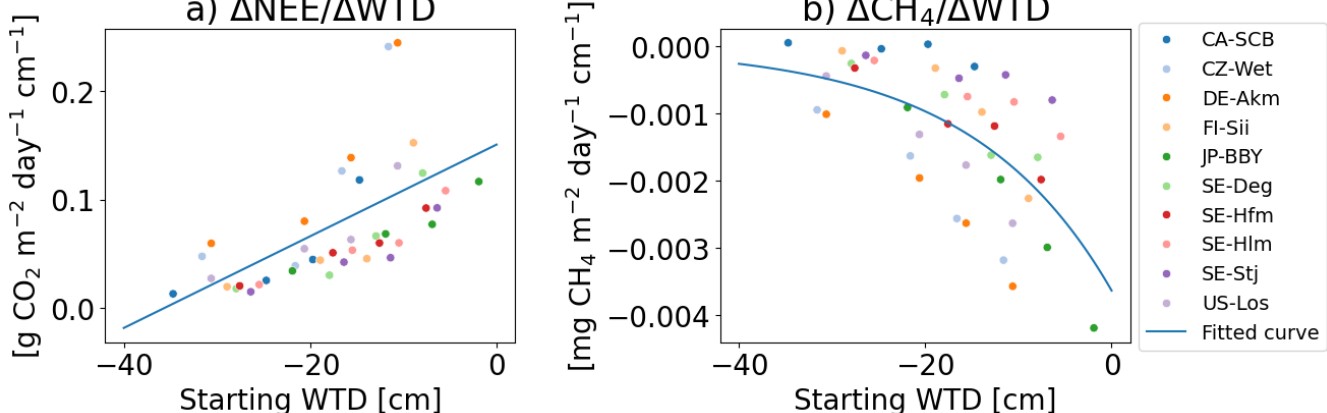

**Figure 7. Sensitivities of NEE and $CH_4$ emissions to water table depth as a function of starting water table depth. The starting water table depth is the upper value in ΔWTD used to calculate sensitivities by dividing Δflux by ΔWTD for all possible negative values of ΔWTD.**


Instead of fitting a linear regression between all the flux changes and all the WTD changes in the simulations (for Fig. 6c,d), we calculated the sensitivity as the change of flux divided by the change of WTD ($\frac{\Delta flux}{\Delta WTD}$) for each possible pair of flux and negative water tables changes, i.e. when WTD becomes deeper, like in Huang et al. (2021b). The 'starting' water table depth in the x-axis of Fig. 7 is the shallower WTD in ΔWTD used to calculate sensitivities. This starting water table level can be
taken from a drained simulation, and thus differs from the 'initial' water table which corresponds to undisturbed conditions. The results shown in Fig. 7 indicate that the more shallow is the starting WTD, the higher are the sensitivities of NEE and $CH_4$ emissions fluxes when the WTD is further lowered. This is because when drainage is applied to a site with a shallow starting water table, it affects upper soil layers that were previously saturated and causes large emissions changes. Conversely, if drainage is applied to a site with a deep starting water table, upper layers that were already exposed to oxygen
experience minimal change and deeper layers dry out but the SOC that they contain decomposes more slowly (they contain more slow and passive carbon with slower turnover time in the model), resulting in a smaller sensitivity of fluxes. The



sensitivities of NEE is a linear function of starting WTD (linear AIC = -532 < non-linear AIC = -530, with AIC definition in Appendix C), while $CH_4$ emissions depends on a non-linear manner of the starting WTD (non-linear AIC = -1351 < linear AIC = -1349 for $CH_4$ emission) and tends to saturate when drainage starts from a deep water table. The sensitivity of $CH_4$ emissions increases from negative values to almost zero when the starting WTD is deeper than $\approx$ 40 cm.

## 3.6 Factors controlling the sensitivity of GHG emissions to water table depth across sites

Figure 8 shows the simulated sensitivities ($\Delta$flux/$\Delta$WTD) of the 10 sites after 10-years of drainage, together with other model variables that could plausibly explain their values. For clarity, the sensitivities are shown in rank from the highest to the lowest for NEE (Fig. 8a). The regression between sensitivities and each of the selected model variables (Fig. 8 c-h) are shown in Fig. 9.

Firstly, we analyzed whether the sensitivities depend on the magnitude of the initial fluxes, based on the hypothesis that a site with a higher baseline flux before drainage would experience more drastic changes during drainage. Among all sites, DE-Akm and CZ-Wet had the highest sensitivities of NEE (0.16 and 0.15 g $CO_2$ m$^{-2}$ day$^{-1}$ cm$^{-1}$, respectively, Fig. 8a). These two sites, along with JP-BB, also exhibited the greatest $CH_4$ emissions sensitivities (-2.24, -2.71, and -3.06 mg $CH_4$ m$^{-2}$ day$^{-1}$ cm$^{-1}$, Fig. 8b). These sites were all characterized by the largest initial $CO_2$ sinks and $CH_4$ emissions before drainage. Conversely, the site with the smallest pre-drainage $CH_4$ emissions, CA-SCB, had the smallest $CH_4$ sensitivities. SE-Stj has the smallest NEE sensitivities with the smallest initial NEE. Overall, we found that the $CH_4$ sensitivities show a very strong linear relationship with initial $CH_4$ emissions ($R^2$ = 0.99, p<0.001) across sites. Similarly, the NEE sensitivities are correlated with the initial NEE, but the relationship is much weaker ($R^2$ = 0.60, p = 0.008) (Fig. 9, top row).





**Figure 8. (a)** Sensitivity of NEE to lowering WTD at each site, defined by the change of each flux per cm of deeper WTD (positive values indicate less $CO_2$ uptake or net emissions with deeper WTD), **(b)** same for $CH_4$ emissions changes (negative values indicate



**less emissions with a deeper WTD), (c) Initial NEE, (d) initial CH$_4$ emissions prior to drainage. Other modeled variables analyzed in the text for their roles in explaining sensitivities: (e) initial water table depth prior to drainage (negative = below surface), (f) annual mean air temperature, (g) modeled soil organic carbon density (SOC) equal to the sum of active, slow, and passive pools over the whole soil column in the model, (h) peatland vegetation composition including peatland graminoid, shrub, moss.**

Secondly, we examined if the vegetation composition of each site affects the sensitivities when the site is drained. To quantify this response, we calculated a moss index (MI) defined by the ratio $\frac{moss-grass-shrub}{moss+grass+shrub}$ so that it is equal to -1 in absence of moss and to 1 with full moss cover, and found a negative correlation between MI and NEE sensitivity ($R^2 = 0.66$, p<0.001), and a positive correlation between MI and CH$_4$ emission sensitivity ($R^2 = 0.34$, p= 0.02) (Fig. 9a,b, last items). We identified two mechanisms in the model which explain this response of sensitivities to the moss fraction. The first

mechanism is that a higher moss fraction is associated with less decomposable organic carbon. Among the three types of peatland vegetation, soil organic carbon decomposes the slowest for mosses and the fastest for shrubs, with residence times of 2 years for moss, 1 year for grass/sedges, and 200 days only for shrubs at >30 °C, for the active soil organic carbon pool of the model. Accordingly, we found that the sites with higher fractions of shrubs and lower fractions of mosses display higher sensitivities of NEE to drainage (Fig. 8). The methane emission sensitivities depend on vegetation cover in a different

way. Peatland sites with mosses have shallower effective root depths (1-5 cm) reducing plant-mediated methane transport in the model, and the opposite is true for grasses/sedges and shrubs (root depth ≈ 30 - 50cm). Most of the sites in this study have an initial mean WTD already below 5 cm, i.e. deeper than the depth at which mosses typically facilitate methane transport. When the WTD gets deeper, plant-mediated transport of methane by moss is thus not strongly affected in our model. Sites with more mosses (e.g. CA-SCB) therefore show smaller methane emission sensitivities. The second

mechanism is that the moss fraction regulates the initial amount of SOC, which partly explains the differences in sensitivities shown in Fig. 8. Note as well that moss dominated sites have a higher potential for oxidation of CH$_4$ due to a symbiosis between mosses and methanotrophic bacteria (Larmola et al., 2010), but our methanotrophy module does not simulate this effect. Sites dominated by shrubs and grasses/sedges (DE-Akm, CZ-Wet) accumulate more SOC before drainage. When drainage exposes the upper peat soil layers to oxygen, accelerating soil respiration and limiting methane production, these

sites with larger initial SOC pools show larger increase in CO$_2$ emissions and larger decrease of CH$_4$ emissions compared to moss dominated sites (CA-SCB, FI-Sii).

Thirdly, we analyzed how the sensitivities depend on the initial WTD across all the sites. We found that the higher (shallower) the initial water table, the higher the sensitivities of NEE and CH$_4$ emissions. This result is qualitatively

consistent with the data shown in Fig. 7 where all sites are displayed together, but it additionally indicates that the site to site differences of sensitivities are partly explained by the variability of initial water table values ($R^2 = 0.60$, p<0.001 for NEE; $R^2 = 0.49$, p=0.03 for CH$_4$ emissions) (Figure 9a,b, middle left items). The initial WTD controls which soil layers are





affected by drainage, which is in turn related to the amount and lability (active, slow, passive) of SOC exposed to oxygen, thereby influencing the sensitivities.


Finally, air temperature was also found to have a correlation with sensitivities. Sites in warmer climates tend to have more sensitive NEE ($R^2$=0.76, p=0.001) and $CH_4$ emission ($R^2$=0.78, p<0.001) responses (Figure 9a,b, middle right items). However, it remains uncertain whether these correlations are causal or reflect co-variations between temperature and other factors, given the small number of points in the regressions shown in Fig. 9.


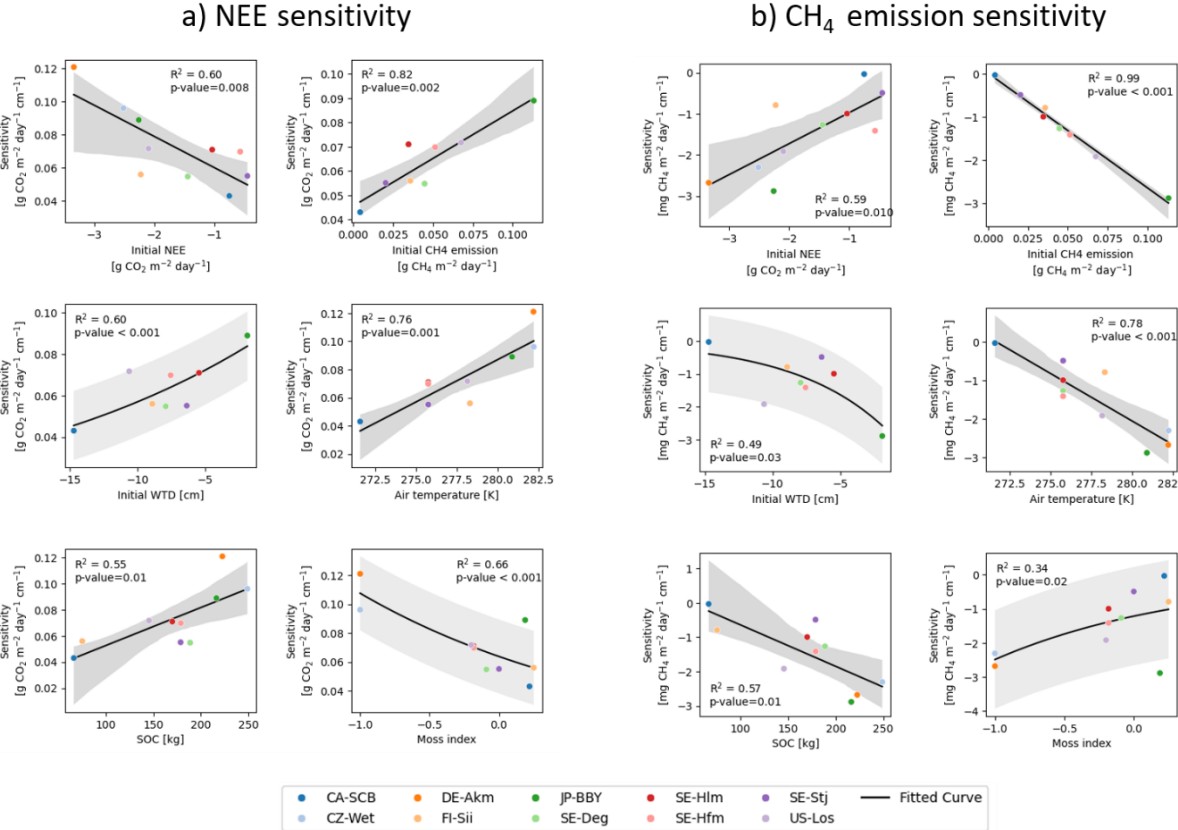

**Figure 9: Regressions between the sensitivities of (a) Net Ecosystem Exchange and (b) $CH_4$ emissions and different model variables including initial NEE, initial $CH_4$ emissions, initial WTD, air temperature, soil carbon content (SOC) and the moss index**

**(MI=$\frac{moss - grass - shrub}{moss + grass + shrub}$).**





### 3.7 Building an emulator of sensitivities for use in GHG accounting and decision support tools.

It is impractical to run a complex, process-calibrated model like ORCHIDEE-PEAT for calculating the sensitivities of GHG fluxes to drainage over a whole region, for instance to derive spatially variables emission factors that could be used in GHG

accounting studies (IPCC Guidelines Tier 3, Eggleston, (2006)) and decision support tools or meta-models like (e.g. FAO, 2021). Therefore, following the analysis of the driving factors of the sensitivities in the previous section (Fig. 9), we developed an emulator of the model sensitivities based on multilinear regressions. To do so, we first fitted all possible regressions of the simulated sensitivities ($S_{NEE}$ and $S_{CH4}$) against the six variables as shown in Fig 9, including initial NEE ($NEE_{init}$), initial $CH_4$ emission ($CH4_{init}$), initial water table ($WTD_{init}$ in cm), soil organic carbon content (SOC in kgC/m$^2$),

air temperature ($T$ in K), and moss index ($MI$). Then, the best regression model was selected based on the minimum corrected Akaike information criterion (AICc, Appendix C), constrained by the cutoff of variance inflation factor (VIF) > 5 to avoid multicollinearity among predictors (Fig. C3):

$$S_{NEE} = 0.045 + 0.406 \times CH4_{init} - 0.020 \times MI$$


($R^2 = 0.91, p <= 0.001$)   (3)

$$S_{CH4} = -13.914 - 28.8991 \times CH4_{init} + 0.051 \times T$$

($R^2 = 0.98, p <= 0.001$)   (4)


To evaluate the robustness of the selected model, a leave-one-out cross-validation was performed. The model was re-fitted multiple times, each time excluding one site from the data, and the performance metrics ($R^2$ and p-value) were recorded for the withheld site (Table C1). The small variation in $R^2$ (+std of $R^2$) and p-value (+std of p-value) among all re-fitted models confirms the robustness of the selected best model.


Furthermore, we calculated the product of the regression coefficients for each variable in Equations (3) and (4) by their standard deviations between sites to quantify the 'effect' of each variable on the sensitivity (Jung et al., 2017). For the sensitivity of NEE, the effect of $CH4_{init}$ is 0.015, higher than that of $MI$, at -0.009. For the sensitivity of $CH_4$ emissions, the effect of $CH4_{init}$ is outstanding (-1.098), four times more important than the $T$ (0.173).

### 3.8 Combined effect of $CO_2$ and $CH_4$ changes







**Figure 10. Change in fluxes (relative to undrained) expressed as CO₂-equivalents by drainage level (x-axis), averaged over 20 and 50 years following the start of drainage. CH₄ emissions were converted to CO₂-equivalents using GWP20 for 20-year average (orange lines) and GWP100 for the 50-year average (green line). In the plot with all sites, the error bars show standard deviations across all sites.**

To assess the combined climate forcing of CO₂ and CH₄ flux changes due to drainage, we used the Global Warming Potential (GWP) metric to convert methane emissions into CO₂-equivalents (IPCC AR6). Figure 10 shows the resulting CO₂-equivalent flux changes averaged across the 10 sites for 20- and 50-year periods of drainage as a function of the WTD drawdown. We used GWP100 = 27 for the 50-year period and GWP20 = 79.7 (IPCC, 2023) shown in different colors in Figure 10.

Averaging our model results for all the 10 sites at 50-cm drainage leads to a slight cooling of -0.11 g CO₂-eq m⁻² day⁻¹ using GWP100 after 50 years of drainage, and to a cooling of -2.02 g CO₂-eq m⁻² day⁻¹ using GWP20 after 20 years. In comparison, from the data collected by meta-analysis studies (Huang et al., 2021b; Zou et al., 2022), retaining only sites where both ΔNEE and CH₄ flux are available, we calculated for all sites a mean cooling effect of -5.46 g CO₂-eq m⁻² day⁻¹ with a huge



spread (95%CI: -73.06 to 17.64) using GWP20, and a mean cooling of -0.06 g $CO_{2-eq}$ $m^{-2}$ $day^{-1}$ (-25.60 to 11.82) using

GWP100, with the occurrence of cooling and warming effects being almost equal among all the sites (Fig. C4). This result is
rather consistent with our model simulations. In contrast, pooling all sites and WTD levels together, thus combining $\Delta CO_2$
and $\Delta CH_4$ from different locations in an inconsistent manner, from the data of Zou et al. (2022) excluding their reported $N_2O$
emissions changes, a net cooling effect of -0.30 g $CO_{2-eq}$ $m^{-2}$ $day^{-1}$ using GWP100 was estimated. Similarly, using a GWP100
value of 25, Huang et al. (2021b) estimated across their sites a net warming effect of 0.033 g CO2-eq/m²/h (0.009 to 0.057).
This suggests that the averaging of sites from meta-analysis data can lead to assessing either a cooling or a warming,

depending on whether only sites with both measured $\Delta CO_2$ and $\Delta CH_4$ are used, or all sites are used.

Our model simulations also show that different drainage periods and the choice of a different GWP time horizon lead to
distinct warming or cooling effects (Fig. 10). For 9 sites out of 10, however, we simulated a larger cooling on a 20-year
horizon compared to a 100-year horizon, due to the stronger radiative forcing impact of reduced $CH_4$ emissions in the short

term. In addition, most of the sites show a larger radiative forcing change when the WTD is deeper, specifically a higher
cooling from reduced $CH_4$ emissions on a 20-year horizon. Secondly, the magnitude and the sign of changes, whether
warming or cooling, varies significantly between sites. CA-SCB is the only site having warming effects (from 0.4 to 1.5 g
$CO_{2-eq}$ $m^{-2}$ $day^{-1}$) regardless of time scales and drainage levels. On the other hand, DE-Akm, JP-BBY, SE-Hfm, and US-Los
almost always have cooling effects (down to -6 g $CO_{2-eq}$ $m^{-2}$ $day^{-1}$) because their $CH_4$ emissions are reduced more than their

$CO_2$ emissions are increased. Other sites have both warming and cooling effects depending on time scale considered and
drainage level. Neutral effects on climate were observed in some cases, primarily when considering 50 years of drainage
(green line), such as FI-Sii with a 10 cm drainage and SE-Deg at a 50 cm drainage level.

## 4 Discussion

Uncertainty in our simulations involves several factors, typically coming from model calibration, our WTD reconstruction,

and the omission of vegetation change during drainage and possible changes of soil structure including compaction.

Firstly, three sites could not be calibrated with $CH_4$ emission but only with NEE, due to a lack of $CH_4$ measurements. A good
simulation of methane processes is expected to improve $CO_2$ emissions because the oxidation of methane in soil produces
$CO_2$ which contributes to the total NEE ($CO_2$ from methanotrophy constitutes 8.21% (95%CI: 0.57% - 29.81%) of

heterotrophic respiration). For sites without $CH_4$ calibration, $CH_4$ parameters were currently taken from the average of
calibrated parameters of other sites. Another solution that could be tried is to do multi-site calibration of $CH_4$ and NEE for
the sites with both observations where they are available , i.e. optimize a single set of parameters that best compromises
between all sites, then take the resulting $CH_4$ parameters for the sites that are not $CH_4$-calibrated. Anyway, uncertainties still
remain, as climate and environmental conditions vary between sites. (Liu et al., submitted) showed that even multi-site





calibration sometimes cannot significantly improve the performance of the model due to the inability of the model to consider all trait-climate correlations, with constant parameters used for traits instead.

Secondly, although we believe that our method to reconstruct WTD from soil moisture using a machine learning (ML) module is an interesting and viable solution for now, the training of a ML model to derive a function from few sites which is
then applied to other sites where WTD is missing could be problematic if extrapolation is required out of training range. We assumed that all peatland sites in this study share the same relationship between soil moisture and water table. However, peatlands in different locations can have different peat soil characteristics, along with varying environmental, terrain, and water supply conditions, so the assumption can inevitably be wrong in some places. One thing we can expect in the future is that when we have more water table data everywhere, this indirect reconstruction of WTD can work better.


Thirdly, changes in vegetation cover on peat soil caused by drainage has not been considered in this study due to the complexity of processes and lack of long term data. With water table drawdown and subsequent physical and chemical alterations, peat soils can become more favorable for certain plants (e.g., vascular plants) and less favorable for others (e.g., some *Sphagnum* mosses), potentially resulting in a shift in vegetation composition (Antala et al., 2022; Kokkonen et al.,
2022). Peatland specialist species adapted to waterlogged conditions may even disappear after water table drawdown (Jassey et al., 2018), while woody species can invade newly available growing spaces (Kokkonen et al., 2019). Additionally, species turnover also depends on the type of peatland (e.g. bog/fen), particularly on hydrological and nutrient conditions prior to drainage (Kokkonen et al., 2022). Changes in plant composition should be incorporated into drainage simulations, as each PFT has unique photosynthetic capacity, respiration rates, and contributes litter of varying quality for decomposition, all of
which would impact the carbon balance. While it's straightforward to introduce in a model a new vegetation composition, simulating the dynamic changes in vegetation - such as determining the factors controlling moss disappearance and adjusting PFT fractions accordingly - remains challenging. Similarly, the model parameters, which were calibrated based on undisturbed fluxes, remained unchanged during the simulation of drainage scenarios, even though soil hydrological parameters such as water holding capacity and water conductivity are known to be affected by compaction during drainage.


In the ditch drainage simulation, we assumed an ideal condition in which the drained peatland is effectively isolated from adjoining land and suppressed the runoff from other soil tiles to the peat soil tile. This is not always what happens in reality, there can still be runoff entering the peatland from adjoining land. However, this suppression helps to make the peat soil effectively dried in the model. With the limitation in the current drainage model's complexity, the ditch drains water from
the peat soil tile only, and not yet from other soil tiles. The runoff from other soil tiles to the peat soil tile that is suppressed in the model can be interpreted as accounting for the water from these tiles that should have also been drained by the ditch. We assumed also a uniform water table drawdown across the whole peatland and over the course of 50 years. However, in reality, the water table near the ditch can differ significantly, or to some extent, from the water table farther away from it.





This depends on the hydraulic conductivity of peat materials, which varies with the degree of peat decomposition (Boelter,
1972). Also, a peatland over time will often recover back to the same water table depth after drainage, as subsidence and
oxidation will bring the surface down close to the previous water table level (Hilbert et al., 2000; Waddington et al., 2015).
Considering such distance and subsidence effects would help move towards a more realistic simulation.

In the analysis of emission factors (EF), we explained their variation by a few processes related to litter and SOC. There are
also other factors that can affect the change of EF by time (whether increasing or decreasing) but were not simulated or
considered in analysis. For example, peat bulk density and porosity can alter due to drainage. Investigating their behaviors
could provide better insights into EF variations. However, in the current ORCHIDEE model, certain soil characteristics (e.g.
peat bulk density and porosity) are treated as constant. Simulating these characteristics as dynamic variables would require
further model development.


We also need in future studies to take into account the emission of $CH_4$ that is dissolved in water drained out, produced in-
situ in the ditch, and diffused at the vertical cross-section surface of the ditch. According to Roulet and Moore (1995), a
drainage ditch is a source of methane because (1) evacuated water transports $CH_4$ from the surrounding peat into the ditch,
and (2) $CH_4$ production is favorable in the ditch where sediment is constantly saturated and warmed by direct solar radiation.
Recent studies based on meta-analysis report an offset of drainage ditch to the methane reduction due to drainage, which is
18% by Peacock et al. (2021)), 12 (10-14) % by Gan et al. (2024)). Roulet and Moore (1995) suggested that flow rate, depth,
and morphology of the ditch can have an impact on its $CH_4$ emission, and ditch spacing also plays a role in net $CH_4$ emission
of the landscape.

Finally, in our simulation protocol, the present climate conditions are recirculated over the whole time of simulation, but it is
interesting for future studies to use climate conditions predicted for the future using climate change scenarios, for instance,
with the Inter-Sectoral Impact Model Intercomparison Project (ISIMIP) database (Lange and Büchner, 2021). Using future
climate forcing is simple, but then we will also need a corresponding forecast of climate induced future WTD changes for
the baseline scenario, and here again our machine-learning model to compute undisturbed WTD from modeled soil moisture
may not work for unseen future soil moisture conditions.

## 5 Conclusions

In this study, we addressed the difficult problem of estimating changes of $CO_2$ and $CH_4$ fluxes when peatlands are drained in
future. Insofar, most of the knowledge comes from empirical results that showed a general increase of $CO_2$ emissions and a
decrease of $CH_4$ emissions during drainage. Yet, meta-analysis data group fluxes from different experiments, where $CO_2$ and
$CH_4$ fluxes are not always measured simultaneously, and include different control conditions, thus showing a large spread of



their results. We used a process model calibrated to match fluxes under pristine conditions before drainage from 10 sites individually and parameterized virtual drainage at each site with a new ditch module. The model was integrated forward for virtual drainage simulations at each site under present climate with different prescribed water table depths. A summary of the answers to the research questions posed in the introduction is given below.

(1) What are the changes of $CO_2$ and $CH_4$ fluxes in response to drainage and how do they compare with observations? We found an increase of $CO_2$ emissions or a decrease of $CO_2$ sinks and a decrease of $CH_4$ emissions from drainage, with a magnitude very similar to the observations in flux data collected by Evans et al. (2021) and within the range of meta-analysis results, although meta-analysis results have a huge variability. On average, we predict for the first decade of 50 cm drainage a reduction of the $CO_2$ sink of 3.14 g $CO_2$ m$^{-2}$ day$^{-1}$ (0.23 to 10.97) and a decrease of $CH_4$ emissions of -0.06 g $CH_4$ m$^{-2}$ day$^{-1}$

(-0.21 to 0.001).

(2) How do fluxes change as a function of drainage duration? We found that a longer drainage period leads to a diminishment of the $CO_2$ emissions increase compared to undisturbed conditions, and to a strengthening of the $CH_4$ emission reduction over time. The first result for $CO_2$ is consistent with previous emission factors simulated for peat drained to croplands agriculture (Qiu et al., 2021). Such a model as presented here, if validated against real-world observations, can

help provide time-dependent emission factors that may be useful for inventory calculations in the absence of long-term $CH_4$ emission factor changes from measurements.

(3) What is the modeled sensitivity of flux changes to water table depth? We found that the shallower the starting water table, the more positive the sensitivity of $CO_2$ flux changes to WTD (more warming) and the greater is the sensitivity of $CH_4$ reductions (more cooling from suppressed $CH_4$ emissions).

(4) What factors affect the sensitivity at each site in the model ? We found that, in the model world, the initial fluxes, initial water table depth, soil organic carbon, moss fraction and temperature are the key influential factors controlling the sensitivities across sites. These variables have strong covariations in the model, so their effect cannot be isolated individually. This finding allowed us to propose an emulator of the modeled sensitivities that could be used to predict flux changes at other sites or over a region. Yet, this is only a model result and testing it against observations would be important

in the future to use such an emulator approach for GHG accounting.

(5) What is the net climate effect of $CO_2$ and $CH_4$ flux changes induced by drainage using the GWP metrics to compare $CH_4$ with $CO_2$. Here we found that averaged over all WTD depths, drainage during 20 years with radiative forcing calculated with GWP20 induces a net cooling because the reduction of $CH_4$ emissions dominates over changes of $CO_2$ fluxes. Drainage during 50 years with GWP100 induces almost a net neutral effect. There is a large variability between sites even for the sign

of the climate effect of drainage. This gives a more nuanced view than the current paradigm that drainage always warms the climate. This result seems at first glance opposite to meta-analysis results, even though our model simulations for flux changes were found to be consistent with these data. The data do not have very long drainage experiments which may explain the differences with our predictions. However, when taking only sites from meta-analysis studies where both $CO_2$ and $CH_4$ fluxes changes were measured, a net cooling is found, similar to our predictions.



Future work should consider additional CH₄ emissions from ditch water and shifts of vegetation composition during drainage. This model framework can be applied for regional historical simulations to improve on previous studies where drainage was not explicitly modeled but represented as an abrupt land cover change from peat to cropland, and where CH₄ effects were ignored. It can also be applied to future rewetting scenarios of degraded peat to assess the net climate effect of this nature based solution.

**Acknowledgements**

This research was carried in the framework of the European Union's Horizon Europe programme WET HORIZONS, grant agreement no. 101056848. E. Salmon is funded by the European Union's Horizon research and innovation program, HORIZON-CL5-2021-D1-01, GreenFeedback under grant agreement no. 101056921.

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

**Appendix A**


**Table A1: The 12 peatland sites used in this study for virtual drainage simulations. Their locations were mapped in Fig. A1. A star (*) beside a site ID indicates flux and WTD observations sourced from the FLUXNET-CH4 database; otherwise, these observations are from the Wet Horizons project. The vegetation composition (graminoids, shrubs, and mosses) is taken from the literature when available; otherwise, it is estimated visually from satellite images.**

| Site ID | Lat, lon | Type | Peat-graminoids/shrub/moss fraction | $CO_2$ flux observation | $CH_4$ flux observation | WTD observation | References |
|---------|----------|------|-------------------------------------|-------------------------|-------------------------|------------------|------------|
| | | | | | | | |



| CA-SCB* | 61.31, -121.30 | Bog, pristine | 0.2 / 0.2 / 0.6 | 2014-2017 | | Yes | (Oehri et al., 2022) |
|---------|----------------|---------------|-----------------|-----------|------|-----|----------------------|
| CZ-Wet | 49.02, 14.77 | Fen, pristine | 0.8 / 0.2 / 0 | 2021-2022 | None | No | (Mejdová et al., 2021) |
| DE-Akm | 53.87, 13.68 | Fen, near-natural | 0.5 / 0.5 / 0 | 2009-2014 | None | No | (Bernhofer et al., 2009) |
| FI-Sii | 61.83, 24.19 | Fen, pristine | 0.26 / 0.1 / 0.64 | 2018-2021 | | Yes | (Aurela et al., 2007) |
| JP-BBY* | 43.32, 141.81 | Bog, pristine | 0.2 / 0.2 / 0.6 | 2015-2018 | | Yes | (Ueyama et al., 2020) |
| SE-Deg | 64.18, 19.56 | Fen, pristine | 0.44 / 0.11 / 0.45 | 2020-2022 | | Yes | (Noumonvi et al., 2023) |
| SE-Hfm | 64.16, 19.55 | Fen, pristine | 0.5 / 0.07 / 0.43 | 2020-2022 | | Yes | (Noumonvi et al., 2023) |
| SE-Hlm | 64.16, 19.57 | Fen, pristine | 0.36 / 0.21 / 0.43 | 2020-2022 | | Yes | (Noumonvi et al., 2023) |
| SE-Stj | 64.17, 19.56 | Fen, pristine | 0.37 / 0.15 / 0.48 | 2020-2022 | | Yes | (Noumonvi et al., 2023) |
| US-Los* | 46.08, -89.98 | Fen, pristine | 0.5 / 0.1 / 0.4 | 2014-2018 | | Yes | (Desai and Thom, 2020) |


.





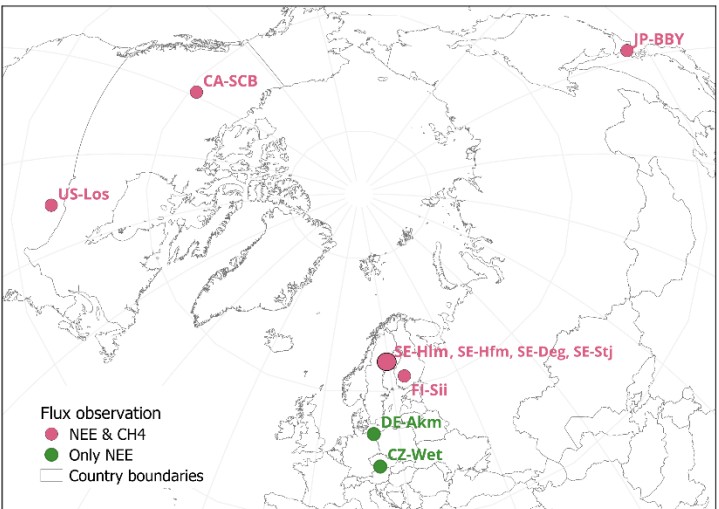

**Figure A1. Location of peatland sites. Figure contains public sector information licensed under the Open Government Licence v3.0.**





**Appendix B**

**Table B1. Parameters for model calibration. The PFT column uses G, S, and M to denote graminoids, shrubs, and mosses, respectively, and "-" for parameters that are independent of PFTs. For further details, refer to (Liu et al., submitted) for parameters related to photosynthesis, autotrophic respiration, and SOC decomposition, and to (Salmon et al., 2022) for methane-related processes.**

| Parameter | Description | PFT |
|---|---|---|
| *Photosynthesis* | | |
| $VC_{max}$ | Maximum rate of carboxylation | G, S, M |
| $LAI_{max}$ | Maximum leaf area index | G, S, M |
| $SLA$ | Specific leaf area | G, S, M |
| $g_0$ | Stomatal conductance of mosses when no irradiance | M |
| $a_1$ | Empirical constants | M |
| $b_1$ | Empirical constants | M |
| *Autotrophic respiration* | | |
| $C_{0,leaf}$ | Maintenance respiration coefficient at 0 °C for leaves | G, S, M |
| $GR_{frac}$ | Fraction of biomass allocated to growth respiration | G, S, M |
| *SOC decomposition* | | |
| $T_{peat}$ | Carbon decomposition rate parameter for peat vegetation | G, S, M |
| $Q_{10}$ | Temperature sensitivity coefficient of the decomposition rate | - |
| *Methane-related processes* | | |
| $q_{MG}$ | Ratio of soil oxic and anoxic decomposition | - |
| $k_{MT}$ | Methanotrophy rate | - |
| $M_{rox}$ | Root methane oxidation | G, S, M |
| $Z_{root}$ | Root depth | G, S, M |
| $T_{veg}$ | The efficiency of methane plant mediated transport | G, S, M |





| Parameter | Description | PFT |
|---|---|---|
| **wsize** | Connectivity of soil moisture | - |
| **mxr$_{CH4}$** | Methane mixing ratio in bubbles | - |
| **O2m** | Oxygen concentration below which anoxic condition is reached for methane production | - |





**Appendix C**

**Text C1. Akaike information criterion (AIC).**

The AIC (Akaike, 1974) is a statistical metric used to evaluate the relative quality of different models for a given dataset by estimating the trade-off between the goodness of fit and model complexity, with lower AIC values indicating a better trade-off. For each model, AIC is calculated by:

$$AIC = 2K - 2ln(L) \tag{C1}$$

where $K$ is the number of estimated parameters in the model and $L$ is the maximum value of the likelihood function for the model, reflecting how well the model fits the data.

The corrected Akaike information criterion (AICc) adjusts the standard AIC to account for small sample sizes (e.g. 10 samples in Sect. 3.7 of this study), ensuring better reliability of model comparisons when the sample size ($n$) is relatively
low compared to the number of model parameters ($K$). The formula for AICc is:

$$AICc = AIC + \frac{2K\,(K+1)}{n-K-1} \tag{C2}$$

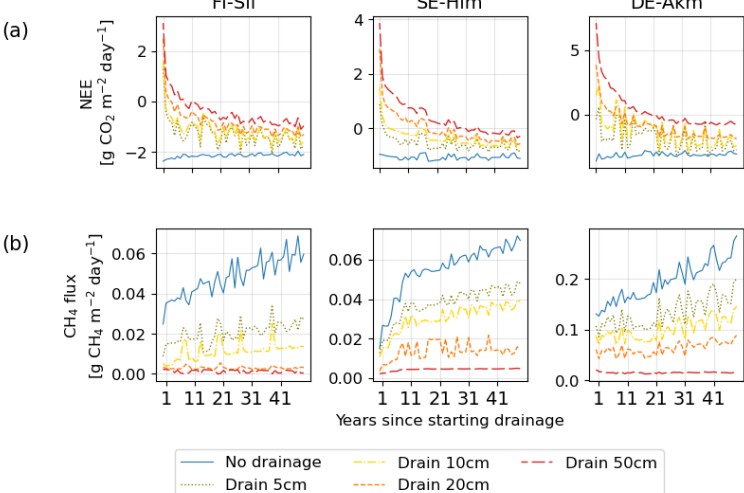

**Figure C0. 50-year time series of (a) Net Ecosystem Exchange of CO2 and (b) methane emissions at three sites as in Figure 4.**
**Different colors indicate different water table depths below the original water level of each site.**



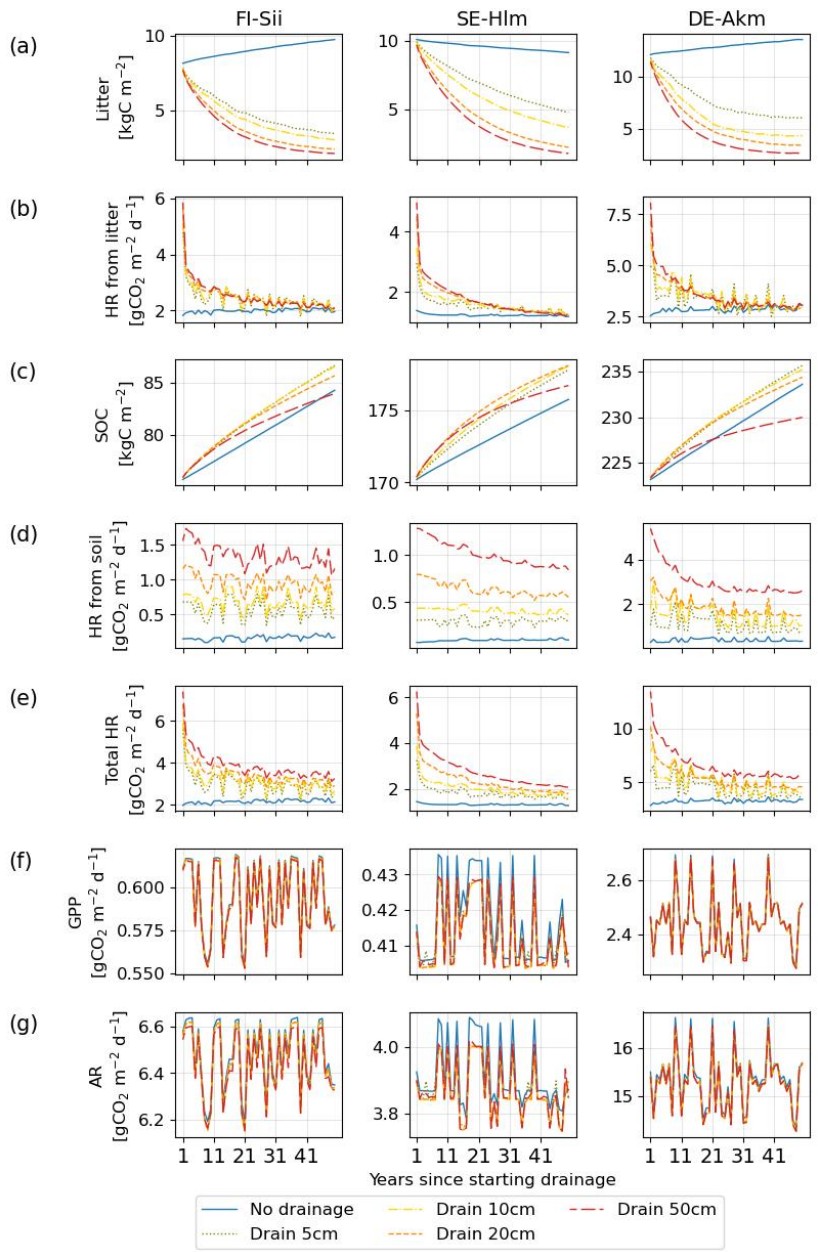

**Figure C1.** Litter and soil organic carbon (a,c) and corresponding heterotrophic respiration (b,d) during the 50-year period of drainage. Total heterotrophic respiration (e) is the sum of that from litter and soil (= b+d).






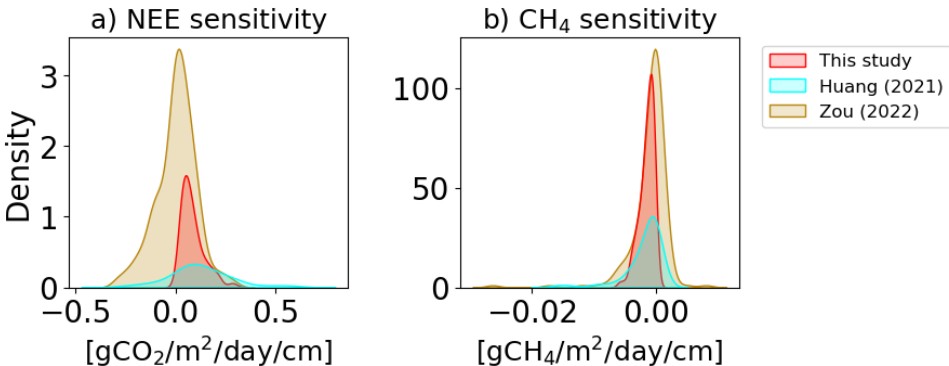

**Figure C2: Distribution of NEE and CH4 emission sensitivity to drainage of peatland sites in this study compared to reference data.**


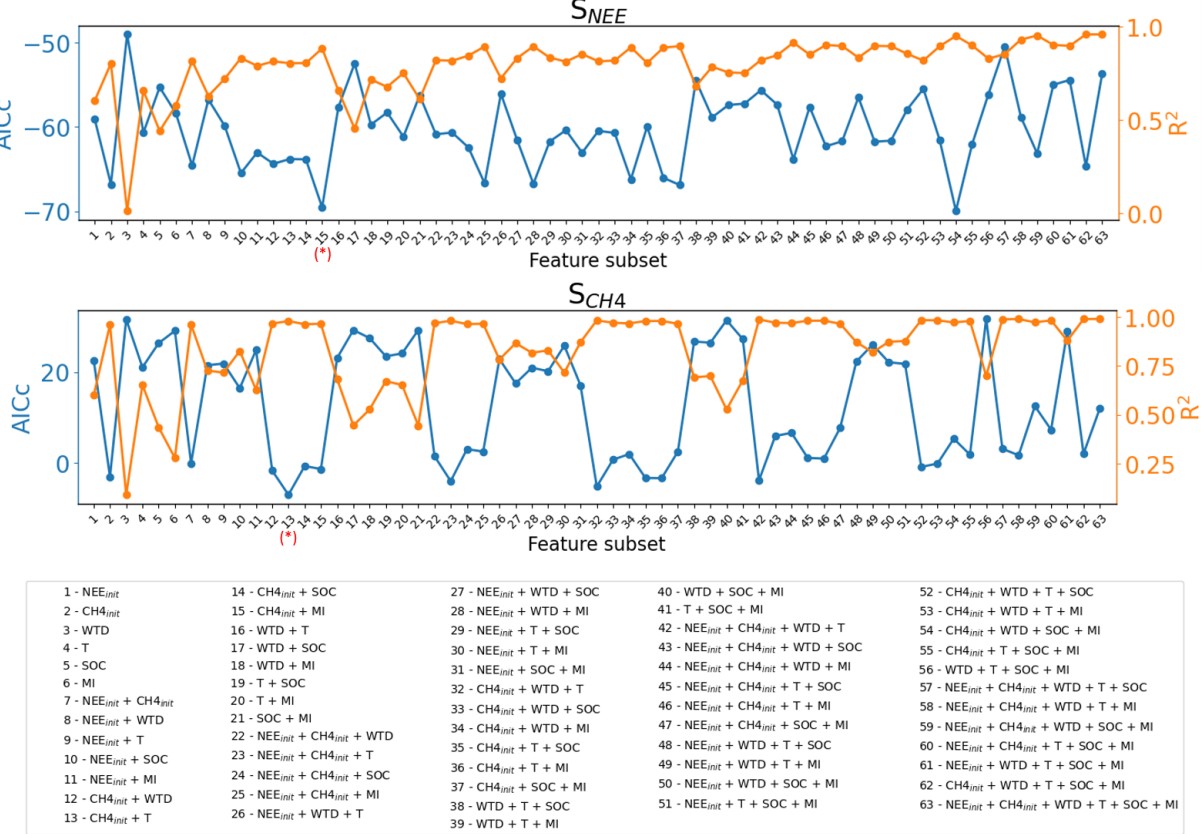

**Figure C3. Selection of controlling variables for flux sensitivities. The red star indicates the subset that was chosen for the emulator (Eq3, 4).**






**Table C1: Coefficient of determination (R²) and p-values from leave-one-out fitting of Equation (3) and (4), with the site removed indicated in the first column.**

| | Fitting Eq (3) | | Fitting Eq (4) | |
|---|---|---|---|---|
| Site removed | $R^2$ | p-value | $R^2$ | p-value |
| CA-SCB | 0.860 | 0.0010 | 0.988 | $10^{-6}$ |
| CZ-Wet | 0.895 | 0.0004 | 0.980 | $10^{-5}$ |
| DE-Akm | 0.889 | 0.0005 | 0.974 | $10^{-5}$ |
| FI-Sii | 0.879 | 0.0006 | 0.980 | $10^{-5}$ |
| JP-BBY | 0.877 | 0.0006 | 0.972 | $10^{-5}$ |
| SE-Deg | 0.897 | 0.0004 | 0.980 | $10^{-5}$ |
| SE-Hfm | 0.885 | 0.0005 | 0.979 | $10^{-5}$ |
| SE-Hlm | 0.906 | 0.0003 | 0.981 | $10^{-6}$ |
| SE-Stj | 0.880 | 0.0006 | 0.978 | $10^{-5}$ |
| US-Los | 0.888 | 0.0002 | 0.984 | $10^{-6}$ |
| Mean ± STD | 0.888 ± 0.013 | 0.0005 ± 0.0002 | 0.980 ± 0.004 | $10^{-5} ± 10^{-6}$ |






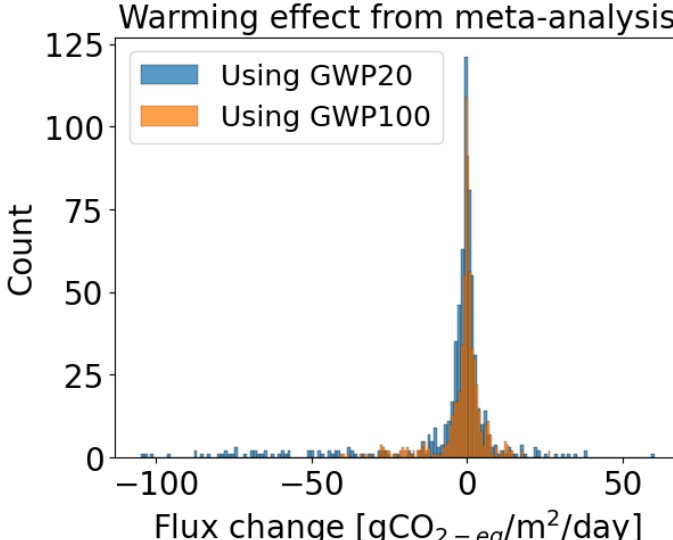

**Figure C4. Distribution of flux change in CO2-eq from meta-analysis data.**