# Peer review of "Modeling the impact of drainage on peatland CO2 and CH4 fluxes and its underlying drivers"

_EGUsphere, 2025_

## Author Comment (AC1)

Summary:

This study calibrates the ORCHIDEE-PEAT model at ten pristine peatland sites to simulate $CO_2$ and $CH_4$ emissions over multiple years under these pristine conditions. Based on the calibrated parameters, the authors conduct hypothetical drainage experiments to investigate how greenhouse gas fluxes change when the water table is lowered. The research aims to address the transitional phase from pristine to drained peatland conditions—a critical phase that is often missing in observational datasets.

The idea is great to fill this knowledge gap with a model-based approach, however, I have serious concerns about how the modeling approach is currently implemented. The key issue is that the calibration is based only on pristine conditions, but the model is used to predict emissions in drained conditions, where apart from water level which is lowered by drainage other environmental controls are fundamentally different. The model does not account for critical processes such as vegetation shifts, changes in soil organic matter, and agricultural practices like biomass removal or fertilization following drainage. These omissions raise fundamental doubts about the reliability of the simulation results and the conclusions drawn by the others.

One highly questionable conclusion of this study is the claim that it provides "a more nuanced view than the current paradigm that drainage always warms the climate." Measurement data indicate that peatland drainage results in long-term climate warming. It is well established that the time horizon is crucial, as the radiative forcing of long-lived greenhouse gases (e.g., $CO_2$) is driven by cumulative emissions, whereas the radiative forcing of short-lived climate forcers (e.g., $CH_4$) depends on contemporary emission rates. Meta-analyses show that, after just one or a few decades, the net impact is unequivocally climate warming. (https://doi.org/10.1038/s41558-019-0615-5, https://doi.org/10.1038/s41467-020-15499-z). Below, I outline my main concerns in more detail.

Thank you very much for your time and effort in reviewing our manuscript, and for your thoughtful comments. We fully acknowledge the limitations highlighted, particularly concerning the calibration under pristine conditions only, and the lack of explicit representation of vegetation shifts, soil organic matter evolution. These are indeed critical processes that, if incorporated, would strengthen the realism of model projections. However, implementing them goes beyond the current capabilities of our model framework and available data. We see this work as a first step toward bridging the gap between idealized pristine conditions and real-world drained peatland dynamics.

Regarding the concern about the conclusion on climate effects, we agree that large-scale assessments and existing literature consistently show a long-term warming effect of peatland drainage. Our intention was not to contradict this view, but rather to highlight that, at the site scale, the net climate impact of drainage can vary depending on site-specific factors such as drainage depth, duration, and initial conditions. Still, we recognize that more robust process representation is needed before drawing generalizable conclusions, and we have revised the manuscript to clarify this and temper the language accordingly.

In the revised manuscript, we (1) added further clarifications and explanations to improve the manuscript based on reviewers' feedback, (2) included a sensitivity experiment to assess the potential impact of vegetation change, and (3) introduced an additional scenario of 80 cm drainage.

Please find our responses to each of your comments below.

**Major Comments:**

1) Unjustified extrapolation beyond the calibration range

The model is calibrated using data from pristine peatlands, but then applied to drained conditions, environmental conditions are entirely different. Using the same parameter set for both conditions is a major extrapolation and likely introduces massive errors.

A better approach would involve calibrating the model using both pristine and drained sites. Multi-site calibration covering the full range of conditions would improve the robustness of the model and reduce uncertainty in the predictions. Without such an approach, the results cannot be interpreted in the context of real world situations.

We agree that parameter values may differ between pristine and drained peatland conditions, and that applying a single parameter set across contrasting environmental states introduces uncertainties as we already noted in the manuscript lines 610-615. However, the sites included in our study have not experienced drainage in reality. As such, we lack observational data from drained conditions that would allow for independent calibration under those scenarios. In the absence of such data, we rely on a consistent parameter set to explore the modeled response to hypothetical drainage. We acknowledge this as a limitation and frame the results as exploratory, aimed at understanding potential system behavior rather than producing site-specific predictions. Future work incorporating data from actually drained sites would certainly strengthen model calibration and validation.

2) Lack of realistic vegetation changes

In reality, peatland drainage always leads to significant changes in vegetation. Many drained peatlands are converted into cropland or grassland, fundamentally altering the plant community, biomass inputs, and ecosystem functioning. However, the model assumes that only water levels and soil moisture change, while vegetation remains the same, i.e. pristine vegetation is assumed to persist under drained conditions, which is clearly impossible for many peat-specific species.

The authors acknowledge this as a limitation, which is good, but I believe it is too severe to allow meaningful interpretation of the results. The fact that model outputs match the range of measured data from drained sites does not justify the approach if the model assumes completely different conditions. The fact that net ecosystem exchange (NEE) values fall within observed ranges does not mean the model is capturing the right processes, it could simply be matching for the wrong reasons.

Moreover, the model does not account for biomass removal, which is critical when peatlands are converted to agriculture, nor for the strong impact of nutrients on peat decomposition.

Without incorporating realistic vegetation and land-use related changes, the results cannot be meaningfully compared to real-world drained peatlands.

Our study specifically addresses hydrological drying through ditch drainage, rather than simulating land-use change (e.g., conversion to cropland or grassland). As such, processes such as biomass removal or nutrient inputs associated with agriculture were intentionally not included. The aim was to isolate and understand the consequences of hydrological changes under simplified conditions, not to reproduce the full complexity of drained agricultural peatlands.

We fully acknowledge that drainage often leads to significant vegetation shifts in reality. However, simulating these dynamics accurately remains challenging. While a Dynamic Global Vegetation Model (DGVM) exists within ORCHIDEE, it currently does not represent peat-specific plant functional types (PFTs) or their interactions with peatland hydrology and nutrient dynamics. Developing such capability would be a valuable future direction.

Nevertheless, we added a sensitivity experiment to evaluate the potential impact of vegetation change in the revised manuscript (Section 4 and Appendix D). Two vegetation scenarios were applied: (1) peatland moss was converted to grass and shrub and (2) peatland moss and grass were converted to shrub. The results suggest that in some cases, the overall patterns remain consistent with the unchanged vegetation scenario, while in other cases, the effect of drainage can be reversed from cooling to warming or vice versa.

*In revised manuscript: Appendix D*
*A sensitivity test was conducted to evaluate how vegetation composition changes affect drainage-induced emissions. Two peatland vegetation change scenarios were applied to all sites, except CZ-Wet and DE-Akm, where mosses are absent from the PFT composition:*
*(1) After 10 years of drainage, 50% of moss was replaced by grass and shrub; after 20 years, all moss was fully replaced by grass and shrub.*
*(2) After 10 years of drainage, 50% of moss and grass was replaced by shrub; after 20 years, both moss and grass were entirely replaced by shrub.*
*The combined effects over 50 years using GWP100 (similar to Fig. 10) are shown in Fig. D1. At site CA-SCB, FI-Sii, and JP-BBY, both vegetation shifts had slight impacts - drainage still resulted in warming effects at CA-SCB and FI-Sii, and cooling effects at JP-BBY. The conversion of moss to other peatland PFTs had little impact at SE-Srj and US-Los, but led to significantly increased drainage-induced $CO_2$-equivalent emission in SE-Deg, SE-Hmr, and SE-HfM (moving up from green to blue lines). The conversion of moss and grass to shrub caused more variations at these five sites (orange lines): drainage can cause a cooling effect instead of warming (e.g. at SE-Hmr, SE-Srj) or vice versa (e.g. at SE-Deg, SE- HfM). These results highlight the importance of accounting for vegetation dynamics in future modelling. Changes in $CO_2$ and $CH_4$ fluxes under the vegetation change scenarios (relative to unchanged PFT scenario) are illustrated in Fig. D2 and D3 for the 80 cm drainage case, with the CO2-equivalent emission of the three vegetation scenarios shown in Fig. D4.*

[Figure]

Figure D1. Combined effect of drainage over a 50-year period using GWP100 under three vegetation scenarios: (green) unchanged PFT as used in the main text; (blue) conversion of moss to grass and shrub; and (orange) conversion of moss and grass to shrub.

[Figure]

Figure D2. Changes in GPP, ecosystem respiration, and NEE under two vegetation change scenarios, relative to the unchanged PFT scenario, for the 80 cm drainage case.

[Figure]

Figure D3. Changes in methanogenesis, methane oxidation (methanotrophy), and methane emissions under two vegetation change scenarios, relative to the unchanged PFT scenario, for the 80 cm drainage case.

[Figure]

Figure D4. CO2-equivalent emission under three vegetation change scenarios for the 80 cm drainage case.

**3) Incomplete representation of soil organic matter (SOM) changes**

The study attributes the simulated decline in $CO_2$ emissions over time to the depletion of labile carbon pools. This is not in line with measured datasets and seems to be a model artifact. The model does not seem to account for the fact that SOM composition changes with ongoing decomposition. Decomposed peat is often more vulnerable to further breakdown than less degraded peat. Observational studies indicate that drainage increases the portion of highly decomposed organic matter, which can sustain $CO_2$ emissions over longer periods (https://doi.org/10.1016/j.geoderma.2019.113911). If the model does not represent this change, then the decline in emissions might not be a realistic outcome but rather an artifact of how SOM dynamics are treated.

In ORCHIDEE, SOC decomposition is governed by first-order kinetics and is influenced by temperature and soil moisture. The SOC is partitioned into three conceptual pools: active (labile carbon, residence time of days to years), slow (partially decomposed material, residence time of years to centuries), and passive (stabilized carbon, residence time of hundreds to thousands of years). The transfer of carbon between these pools, along with $CO_2$ release, is handled via fixed turnover time and partitioning coefficients. These turnover times are parameterized during model calibration, and the active pool residence time serves as a reference point for the others.

We acknowledge your point that, in reality, the reactivity of SOM evolves over time, and highly decomposed peat can sometimes be more prone to further decomposition. It is a known limitation of the current model that it does not dynamically represent shifts in SOM quality or microbial activity. It does not explicitly track the progressive degradation of individual SOM components or nutrient feedback. The model only simulates carbon move between pools and decomposition could be calculated based only on the residence time calibrated under pre-drained conditions. We added this point to the revised manuscript (Section 4 - Discussion), but for now we have not been able to upgrade the model to that complex level.

*In revised manuscript: Section 4*
*Additionally, it was found that highly decomposed peat due to drainage can be more vulnerable to further breakdown (Säurich et al., 2019), which is influenced by both peat properties and nutrient status. Yet, our current model does not explicitly track the*

*progressive degradation of individual SOC components or incorporate nutrient feedbacks to diagnose their corresponding reactivity. Decomposition could be calculated based only on a fixed residence time which is a parameter calibrated under pre-drained conditions. Simulating peat properties and their feedbacks as dynamic variables would require further model development.*

**4) Weak validation approach**

The validation method used in the study is not robust enough to assess the model's reliability. The so-called "80/20" split seems to rely on arbitrary data partitioning, possibly in seven-day blocks. Given the strong temporal autocorrelation in peatland flux datasets (also for seven-days), such an approach can lead to overestimated model performance (https://doi.org/10.1111/ecog.02881). More rigorous validation strategies, i.e. a more systematic approach to separating training and validation datasets, should be used. Independent time periods or entirely different sites should be used for validation.

To clarify, the "80/20" validation approach was used exclusively for the water table depth (WTD) reconstruction model and not for the calibration or validation of CO2 or CH4 fluxes. These processes were handled separately and are independent in our study.
We understand the concern regarding temporal autocorrelation potentially inflating model performance. However, in our case, the observational records available for WTD at each site are relatively short (typically only a few years), making it difficult to withhold a full year for validation. We employed a seven-day block sampling strategy instead of selecting individual days, which distributes both training and validation data more evenly across the entire time series. This approach helps reduce the risk of overfitting to short-term fluctuations or seasonal biases and provides a more representative validation across the full observation period.
As for the idea of using completely independent sites for validation, we agree that in principle this is more rigorous. However, WTD-moisture relationships are highly site-specific due to differences in peat properties, vegetation, microtopography, etc. Therefore, using one site to validate another's WTD reconstruction model would likely lead to misleading results due to structural differences between the sites rather than actual model performance.
While we acknowledge the limitations of our approach, we believe that our strategy represents a practical compromise given the short time series available and the spatial heterogeneity among sites.

**5) Unclear rationale for using an ML-based water level model**

Previous studies using ORCHIDEE-PEAT have simulated peatland water levels using process-based approaches. However, in this study, a machine learning (ML)-based model is used instead. The reason for this shift is not well explained, nor is the ML model sufficiently validated. What advantages does this approach offer over a physically based model? Without a strong rationale and proper validation, it is difficult to assess whether the ML approach improves or weakens the reliability of the results. More transparency on this choice would help clarify its impact on the findings.

The WTD simulation in ORCHIDEE-PEAT, which relies on a simplified water budget approach rather than fully mechanistic hydrological processes, can capture broad seasonal trends but often struggles with reproducing site-specific short-term fluctuations. In this study, we had

observed WTD data for 8 out of 10 sites, though some time series were containing gaps. The machine learning (ML)-based model was chosen specifically to take advantage of these observations: it allowed us to reconstruct a continuous and more detailed WTD time series that aligns better with measured data, including finer temporal dynamics.

As noted in the original manuscript (Section 2.2), the numerical discretization of soil layers becomes coarser with depth, which reduces the accuracy of the model's WTD prediction. We have now added further clarification in the revised manuscript (lines 195-196, 197-199) to explain the rationale behind using an ML-based approach in place of the physically based method.

> *In revised manuscript:* Section 2.2
> *Performing drainage simulations requires a good representation of the baseline water table (WTD) before drainage. This is not straightforward in a model like ORCHIDEE where the numerical discretization of the soil into layers is coarser with increasing depth, e.g. a layer has thickness of 25 cm at 50 cm below the surface, which does not allow to position the water table in this layer accurately, **i.e. the model can capture broad seasonal WTD trends, but often struggles to reproduce short-term fluctuation.** Therefore, we developed a machine learning module, separate from the ORCHIDEE model, to simulate the accurate position of the water table as a function of simulated soil moisture in the soil layers, **which leverages the WTD observation data that we have for 8 out of 10 sites. It is expected to reconstruct a continuous and more detailed WTD time series that aligns better with measured data, including finer temporal dynamics.***

**6) Insufficient explanation of the drainage module**

The study mentions the implementation of a drainage module, but it is unclear what exactly this entails. One key issue is whether it realistically represents soil moisture dynamics in the unsaturated zone. Capillary forces play a crucial role in maintaining moisture levels above the water table, and ignoring them can lead to incorrect soil moisture predictions. The paper does not make it clear whether this critical process is accounted for. A more detailed description of how the drainage module works with details on the underlying physics of the soil hydraulic approach would improve clarity.

We respectfully believe that the drainage module was described with sufficient clarity in the manuscript. In ORCHIDEE, vertical water fluxes are simulated only in the downward direction—from the surface layer to lower soil layers. Capillary rise or upward water movement due to capillary forces is not currently represented in the model. This is a known limitation of ORCHIDEE's hydrology scheme. We now added a note in the description (Section 2.3). Anyway, we acknowledge that including capillary processes would improve the realism of soil moisture simulations and agree that future model development should aim to address this limitation.

> *In revised manuscript:* Section 2.3
> *Note that in the ORCHIDEE model, vertical water fluxes are simulated only in the downward direction—from the surface layer to lower soil layers. Upward water movement due to capillary forces is not currently represented in the model.*

Regarding model evaluation, as stated earlier, our study focuses on pristine peatland sites that have never been subject to real drainage. Consequently, no observational data on soil moisture in the unsaturated zone under drained conditions is available for these locations. This makes it impossible to directly validate soil moisture dynamics in the unsaturated zone for drained scenarios.

**Minor Comments**

Line 40: Given the limitations of the study, the claim made here seems overly optimistic. Since the authors themselves acknowledge several key limitations, this statement is sufficiently supported by the study results.

We changed this sentence (Line 40).
*Our model-simulated sensitivities of GHG fluxes to drainage can be approximated by linear regressions using site-level variables, which, despite the study's limitations, may offer a simplified tool for estimating drainage effects.*

Lines 66–70: The text suggests that only "some" studies report long-term $CO_2$ emissions after drainage, but in fact, the vast majority of research supports this finding. Evidence from drained peatlands in the UK, Netherlands and Germany, where meters of peat have been lost (compaction alone cannot explain this) over time evidenced by timber posts (e.g. https://www.greatfen.org.uk/about-great-fen/heritage/holme-fen-posts, many more exist), contradicts any suggestion that emissions decline significantly in the long run if there is still peat available to be oxidized.

We have revised the sentence to better reflect the consensus in the literature regarding long-term $CO_2$ emissions following peatland drainage.
*Meta-analyses consistently suggest that post-drainage peat decomposition causes long-term legacy CO2 emissions, persisting decades after drainage (Couwenberg et al., 2010; Huang et al., 2021b; Zou et al., 2022).*

Line 71: The terminology and sign convention regarding water table level and water table depth is inconsistent throughout the manuscript. This should be standardized to avoid confusion about what high, low, deep, shallow, etc. means.
We corrected them.

Final comments:
I think the focus on the transitional phase between pristine and drained peatland conditions is valuable, as this is an important and underrepresented topic in peatland research. However, the modeling approach has fundamental weaknesses that limit the reliability of the findings.
I hope these comments are taken in the constructive spirit in which they are intended. Improving these aspects in a fundamentally revised paper will help strengthen the study and ensure that its conclusions are robust.

---

## Author Comment (AC2)

Reviewer 1 has provided a thorough and insightful evaluation of the manuscript, highlighting several key issues that warrant attention. I won't repeat all their points here, but I would like to express my full agreement with their assessment.

That said, I also share Reviewer 1's view that the paper addresses an important topic—modeling peatland drainage and its effects on greenhouse gas emissions. This is a valuable area of research that deserves continued development. The modeling approach appears to perform quite well under pristine conditions, especially where the water table depth (WTD) remains above -20 cm in the calibration datasets. In that respect, the paper serves as a strong example of model development. It may also serve to inform of what may happen to pristine systems as these are affected by warmer and drier climates.

However, the system being modeled—drained peat soils—is highly complex. These soils behave quite differently from mineral soils or even undisturbed peatlands. Lessons from better-studied systems may not always transfer well, especially for peatlands that have been drained for long periods.

My primary concern lies in some of the claims made based on this model. These are bold conclusions, and in such cases, it's essential to ensure that they are supported by robust evidence. At present, I'm not fully convinced that the data and analysis sufficiently back these claims.

There are many innovative aspects in the manuscript, but the results hinge critically on the assumption of a slow SOC pool in deeper peat layers. As Reviewer 1 has pointed out (and I had also intended to reference), this assumption may not reflect reality. In fact, evidence suggests that in agricultural peatlands, decomposition rates often increase over time as the peat becomes more degraded. This point should be reconsidered in future model iterations.

Thank you very much for your time and effort in reviewing our manuscript, and for your constructive feedback. We fully agree that the drained peatland system is highly complex and that conclusions based on current model outputs must be interpreted with caution. The model still has many shortcomings and needs to develop more. Our aim was to take an initial step toward simulating how a pristine peatland would respond to drainage, as a first move toward more realistic modeling of drained peatland dynamics, not to draw any broad or definitive conclusions. We have moderated the strength of our conclusions accordingly.

Regarding the SOC decomposition dynamics, we share your concern. As noted in our response to Reviewer 1, the model does not currently simulate progressive changes in SOM quality or associated increases in reactivity due to degradation. Instead, decomposition is governed by fixed turnover times across conceptual carbon pools. We recognize this as a key limitation and have added a statement to the Section 4 - Discussion to highlight the need for future model improvements in this area.

*In revised manuscript:*

*Additionally, it was found that highly decomposed peat due to drainage can be more vulnerable to further breakdown (Säurich et al., 2019), which is influenced by both peat properties and nutrient status. Yet, our current model does not explicitly track the progressive degradation of individual SOC components or incorporate nutrient feedbacks to diagnose their corresponding reactivity. Decomposition could be calculated based only on a fixed residence time which is a parameter calibrated under pre-drained conditions. Simulating peat properties and their feedbacks as dynamic variables would require further model development.*

In the revised manuscript, we (1) added further clarifications and explanations to improve the manuscript based on reviewers' feedback, (2) included a sensitivity experiment to assess the potential impact of vegetation change, and (3) introduced an additional scenario of 80 cm drainage.

I'd also like to raise a few additional considerations:

**1. Drainage depths used in the model runs**: The levels tested (-5, -10, -20, and -50 cm) seem quite shallow. Only the -50 cm scenario might resemble an initial drainage condition. Including deeper, more representative drainage levels could provide more realistic insights.

We added a drainage of 80cm in the revised version. All the figures and statistics were updated accordingly.

**2. Dynamics of managed drainage systems**: In real-world settings, drainage in peat soils is often actively maintained over time—ditches are deepened, compacted soils are re-drained, etc. This means that drainage may reach progressively deeper peat layers, encountering different SOM characteristics than those present at the start. The model could reflect this temporal change more explicitly.

Thank you for highlighting this important aspect of managed drainage systems. While we did acknowledge in the Discussion that subsidence may cause the peat surface to lower over time - bringing it closer to the water table - we did not explicitly consider the common practice of actively maintaining drainage efficiency by deepening ditches or re-draining compacted soils. We agree that this practice can result in progressively deeper drainage over time, potentially exposing peat layers with different SOM characteristics than those initially affected. Capturing such dynamic feedback would require additional model developments, particularly in representing peat structural changes, compaction, and adaptive ditch management strategies. We acknowledge this as a further limitation and have clarified this point in the revised Discussion.

*In revised manuscript:*

*Additionally, in managed systems, drainage is often actively maintained over time - such as by deepening ditches - to preserve drainage efficiency. This practice can lead to a progressive lowering of the effective water table, further exposing deeper peat layers with different SOC characteristics. Incorporating such dynamics would require modelling not only peat subsidence, as already noted, but also adaptive ditch management, which is not yet represented in the current model framework.*

**3. Use of GWP for interpreting results**: While GWP is commonly used in policy contexts, it may not always capture the complexity of peatland carbon dynamics—particularly the long-term $CO_2$ storage versus $CH_4$ emissions trade-off. Radiative forcing models might offer a more nuanced picture and could be used to contextualize the GWP results more effectively.

We totally agree with this opinion. In fact, we were going to do that: using a radiative forcing model to evaluate the net climate impact instead of GWP. However, we found this manuscript already long and did not want it to have more content. So we used GWP for now and considered the run with radiative forcing model as a potential follow-up to this study.

**A few minor suggestions:**

Please clarify whether the point data in the figures represent actual site measurements or model outputs.

It's done.

In line 495, it seems there may be a typo in the equation: should it be intNEE rather than $intCH_4$?

It was indeed the initial CH4. Initial CH4 was found to have a strong relationship with NEE sensitivity.
Anyway, with the 80cm drainage updated, the equation changed now (although initial CH4 is still highly related to the NEE sensitivity).
*In revised manuscript:*

$$S_{NEE} = 0.022 - 0.013\,NEE_{init} - 0.025\,MI \quad (3)$$

*(R2 = 0.92, p< 0.001)*

$$S_{CH4} = 0.045 - 14.426\,CH4_{init} \quad (4)$$

*(R2 = 0.998, p< 0.001)*

With a more realistic representation of drained peat soils, this modeling framework has strong potential to inform both research and policy. I hope these suggestions are helpful and supportive of the manuscript's continued development.